# Training data size induced double descent for denoising neural networks and the role of training noise level.

## Abstract

When training a denoising neural network, we show that more data isn't more beneficial. In fact the generalization error versus number of of training data points is a double descent curve.

Training a network to denoise noisy inputs is the most widely used technique for pre-training deep neural networks. Hence one important question is the effect of scaling the number of training data points. We formalize the question of how many data points should be used by looking at the generalization error for denoising noisy test data. Prior work on computing the generalization error focus on adding noise to target outputs. However, adding noise to the input is more in line with current pre-training practices. In the linear (in the inputs) regime, we provide an asymptotically exact formula for the generalization error for rank 1 data and an approximation for the generalization error for rank $r$ data. We show using our formulas, that the generalization error versus number of data points follows a double descent curve. From this, we derive a formula for the amount of noise that needs to be added to the training data to minimize the denoising error and see that this follows a double descent curve as well.

## 1 Introduction

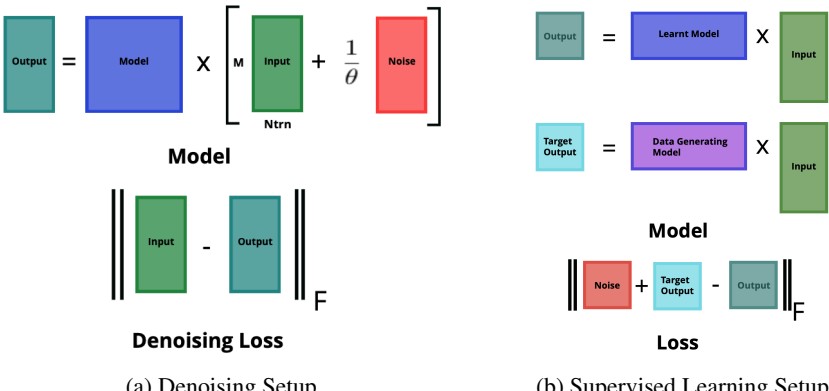

(a) Denoising Setup        (b) Supervised Learning Setup

Figure 1: Figure showing the difference in the noise placement between the traditional supervised learning set up for which empirical and theoretical double descent curves have been found versus our denoising set up for which we recover double descent curves.

Denoising noisy training data is a widely used technique for pretraining networks to learn good representations of the data. Two extremely common examples of pretraining via denoising are Masked Language Modelling (MLM) (Devlin et al., 2019) and Stacked Denoising Autoencoders (SDAE) (Vincent et al., 2010). For many modern problem, we work at large scales in terms of the number of parameters and the number of training samples. Recently there has been significant work in understanding the effect of scaling the number of parameters in the neural network. This resulted

in the discovery of the much celebrated double descent phenomena (Belkin et al., 2019). However, we do not have as good of an understanding of the effect of scaling the number of data points. Works such as Nakkiran et al. (2020); Nakkiran (2020); d'Ascoli et al. (2020); Adlam & Pennington (2020) show either empirically or via theoretical analysis that sample wise double descent exists. However, these were in the regime of supervised learning. On the other hand, our motivation comes from understanding denoising autoencoders. For MLM and SDAEs the denoising is a pretraining procedure, in which case the generalization error would depend on the downstream task. We shall instead look at the generalization error with respect to denoising test data. The difference between prior supervised learning set up and our denoising set up can be seen in Figure 1.

In an attempt to theoretically understand the denoising setting, we look at the simplest setting. Specifically we look at the case, when our network is a linear (with respect to the inputs) network and we are denoising data that lies on a line embedded in high dimensional space. In this setting, we derive the exact asymptotics for the generalization error. We see that in this case, the generalization error spikes at the interpolation threshold (Figure 5a) and the amount of noise that we want to add also spikes at the interpolation threshold (Figure 5b). From the theoretical analysis, we see that the spike occurs due to the variance of the model increasing.

**Contributions.** The main contributions of our works are as follows.

1. We empirically show that when denoising data using a feedforward network, the curve for the generalization error versus the number of training data points as well the curve for the ratio of the test data SNR to the optimal training data SNR has double descent. Further changing the training data SNR can mitigate the double descent in the generalization error curve.
2. Assuming we have mean 0, rotational invariant noise, we derive an analytical formula for the expected mean squared generalization error for denoising rank 1 data by a linear network. Further, we use the same method to derive an approximation for higher rank data and experimentally verify the accuracy of the formula for general low rank data.
3. Using our formula, we show that even in this simple model, we see that the double descent exists for the generalization error and for the amount of noise that should be added versus the number of training data points.

**Related work.** Understanding deep neural networks is current active area of research with many exciting theoretical results. The discovery that fixed depth infinite width neural networks can be thought of as kernel regression (Jacot et al., 2018) and the discovery of double descent for neural networks (Belkin et al., 2019) has sparked significant research into understanding the generalization in the linear regime (in parameters not inputs). The exact asymptotic for generalization loss were first understood for ridge regression (Bartlett et al., 2020; Hastie et al., 2019; Belkin et al., 2020; Advani & Saxe, 2020). This was further generalized to understand the situation for the Random Features model and the Neural Tangent Kernel (NTK) model (Mei & Montanari, 2019; Ghorbani et al., 2019; Adlam & Pennington, 2020). A partial list of recent work for supervised learning includes Jacot et al. (2020); Mel & Ganguli (2021); Derezinski et al. (2020); d'Ascoli et al. (2020); Dobriban & Wager (2015); Geiger et al. (2019); Lampinen & Ganguli (2019); Liang et al. (2020); Muthukumar et al. (2019); Loureiro et al. (2021). However, there has been no work, to our knowledge, that looks at the problem either empirically or theoretically for the denoising set up.

The idea of adding noise to improve generalization has been seen before. One popular strategy is to use Dropout (Hinton et al., 2012; Wan et al., 2013; Srivastava et al., 2014), where we randomly zero out either neurons or connections. Another idea that is commonly used is data augmentation. In a revolutionary paper, Krizhevsky et al. (2012) showed that augmenting the dataset with noisy versions of the images, greatly improved the accuracy. Another area where noise is useful is adversarial learning. Dong et al. (2021) shows epoch wise double descent for adversarial training.

In terms of recent theoretical work related to SDAEs, Pretorius et al. (2018) derived the learning dynamics of a linear autoencoder in the presence of noise. They also establish some relationships between the noise added and weight decay. However, they do not look at the generalization error or quantify the optimal amount of noise that should be added. Gnansambandam & Chan (2020) looked at the problem of what is the optimal amount of noise that should be added. However, they studied this from the perspective of minimizing the variance of the performance.

## 2 SET-UP

Let $U \in \mathbb{R}^{M \times r}$ be our feature matrix. For ease of notation, we assume that the columns of $U$ have unit norm and are pairwise orthogonal. Then to generate $N$ data points, we sample our latent variables $V \in \mathbb{R}^{r \times N}$ and $\Sigma \in \mathbb{R}_+^{r \times r}$ such that $V$ has columns that have unit norm and are pairwise orthogonal and $\Sigma$ is a diagonal matrix such that $\|\Sigma\|_F = 1$. Then a data matrix $X$ is given by $X = U \Sigma V^T$. For us, we have two matrices $X_{trn}$ and $X_{tst}$ that correspond to the train and test data sets. Hence we have corresponding $V_{trn}^T \in \mathbb{R}^{r \times N_{trn}}$, $V_{tst}^T \in \mathbb{R}^{r \times N_{tst}}$, and $\Sigma_{trn}, \Sigma_{tst}$. We make no other assumptions on $U, V_{trn}, \Sigma_{trn} V_{tst}, \Sigma_{tst}$ except that they are given and fixed. Finally, let $\theta_{tst}, \theta_{trn} \in R_+$ be scalars that will scale the singular values of $X_{trn}, X_{tst}$ so that we can control the SNR. We also assume that $\theta_{tst}$ is fixed and that we have control over $\theta_{trn}$. Let $c = M/N_{trn}$ and let $A_{trn}, A_{tst}$ be noise matrices that are added to the training and the test data. Let $W$ be the linear autoencoder that is the solution to the following problem

$$\text{minimize}_{\hat{W}} \quad \|\theta_{trn} X_{trn} - \hat{W}(\underbrace{\theta_{trn} X_{trn} + A_{trn}}_{Y_{trn}})\|_F^2. \tag{1}$$

Then, the expected mean squared error, is given by

$$R(\theta_{trn}, \theta_{tst}, c, \Sigma_{trn}, \Sigma_{tst}) := \mathbb{E}\left[\frac{\|\theta_{tst} X_{tst} - W(\theta_{tst} X_{tst} + A_{tst})\|_F^2}{N_{tst}}\right]. \tag{2}$$

### 2.1 ASSUMPTIONS ABOUT THE NOISE

We assume that each entry of the noise matrix $A$ has mean 0, variance $1/M$ and that the entries of $A$ are pairwise uncorrelated. Additionally, we shall assume that $A$ is rotationally bi-invariant. That is, if $Q$ is an $M$ by $M$ ($N$ by $N$) orthogonal matrix, then $QA$ ($AQ$) has the same distribution as $A$. Another way to phrase this is if $A = U_A \Sigma_A V_A^T$ is the SVD, then $U_A$ and $V_A$ are uniformly random orthogonal matrices and are independent from $\Sigma_A$ and each other. Finally, we shall assume that $A$ has full rank with probability 1 and that the limiting distribution of the eigenvalues of $A^T A$ converge to the Marchenko-Pastur distribution. While such assumptions on the noise may seem restrictive. This encompasses a large family of noise distributions.

**Proposition 1.** *If $B$ is a random matrix that has full rank with probability 1 and its entries are independent, have mean 0, have variance $1/M$, and bounded fourth moment, and $P, Q$ are uniformly random orthogonal matrices. Then $A = PBQ$ satisfies all of our noise assumptions.*

### 2.2 SIGNAL TO NOISE RATIO (SNR)

A quantity of interest to us will be the SNR, given by $\|X\|_F / \|A\|_F$. Hence, we need to normalize everything by $\|A\|_F$. In this case, due to our assumptions, we have that $\mathbb{E}[\|A\|_F^2] = N$. Hence, for any variables and constants, if it has a hat, then that refers to that variable or constant normalized by $\sqrt{N}$. For example, given $\theta_{trn}, X_{trn}$, and $A_{trn}$, then we have that

$$\|\theta_{trn} X_{trn}\|_F / \|A_{trn}\|_F = \theta_{trn} / \|A_{trn}\|_F \approx \theta_{trn} / \sqrt{N_{trn}} =: \hat{\theta}_{trn}.$$

## 3 EMPIRICAL DOUBLE DESCENT

We first show that sample wise double descent occurs for denoising neural networks empirically. Figure 2 shows that if we train a feedforward network to denoise data such that the training data signal to noise ratio (SNR) $\hat{\theta}_{trn}$ is the same SNR as that of the test data set ($\hat{\theta}_{tst}$), then the curve for the denoising generalization error vs the number of training samples has the shape of a double descent curve. Thus, together with prior work, this suggests that the double descent with respect to the number of data points is a universal phenomena. However, unlike other hyperparameters, such as number of features and number of training epochs, we cannot arbitrarily change the number of data points as we are limited by the data set that we have. Hence it could be the case, that the maximum number of data points that we have corresponds to the peak of the generalization error curve.

However, we can look at the amount of noise that we add to the training data. To see the effect of the noise, for a variety of different $\hat{\theta}_{trn}/\hat{\theta}_{tst}$, we compute the denoising generalization error versus

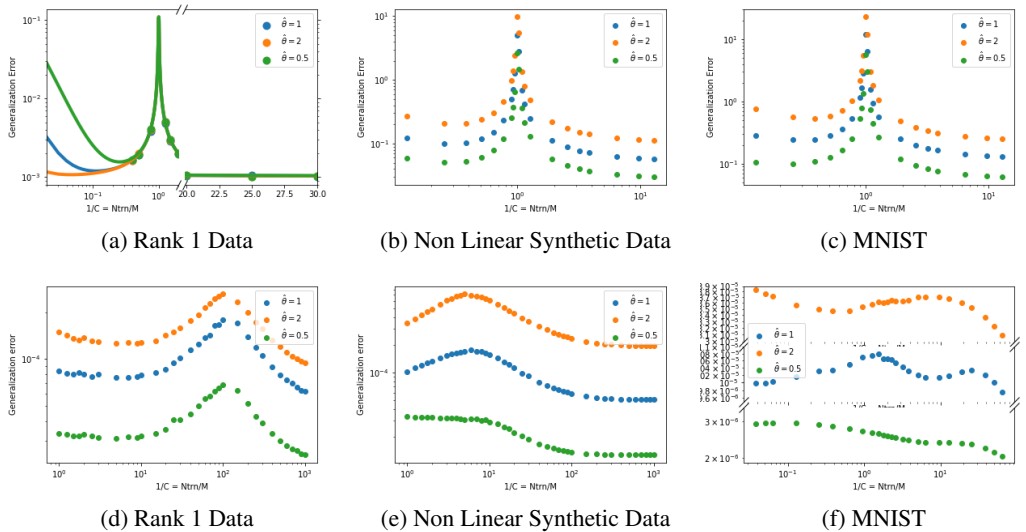

(a) Rank 1 Data      (b) Non Linear Synthetic Data      (c) MNIST

(d) Rank 1 Data      (e) Non Linear Synthetic Data      (f) MNIST

Figure 2: Figure showing the double descent phenomena for the generalization error versus the number of the training data points. The top row is for a linear network and the bottom row is for a 3 layer ReLU network. Here the training data SNR and the test data SNR both equal $\hat{\theta}$.

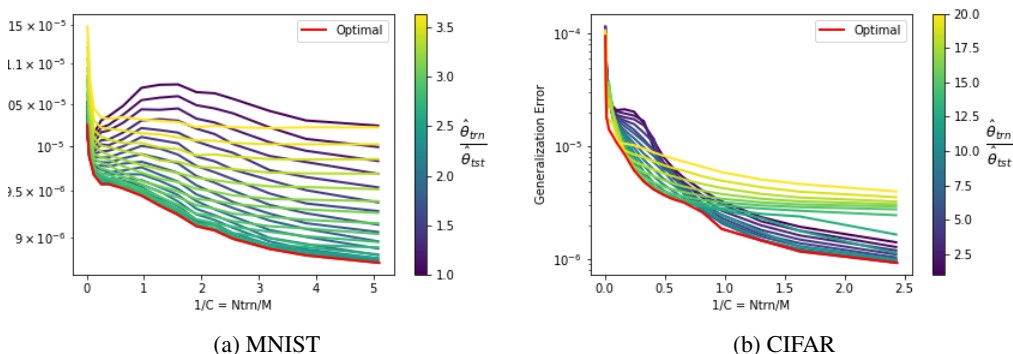

(a) MNIST      (b) CIFAR

Figure 3: Figure showing the denoising generalization error for a 3 layer neural network trained for various different values of $\hat{\theta}_{trn}/\hat{\theta}_{tst}$ and number of training data points. Each neural network was trained for 1500 epochs, using gradient descent with a learning rate of $10^{-3}$. For MNIST, we averaged over 20 trials and for CIFAR10 we averaged over 5 trials.

the number of data points curve. We do this for MNIST and CIFAR dataset. We create test data sets by taking the test data for each and then adding Gaussian noise. We fix the test SNR to be 1 for both datasets. Hence we know the test data SNR. We then take various different fractions of the training data and train a 3 layer ReLU neural network (without bias) for various levels of training noise. For each of pair of parameters (number of training data points and the level of training noise), we compute the generalization error averaged over 20 trials for MNIST and 5 trials for CIFAR. Here the test noise and training noise is resampled for each trial. The plots for the generalization error can be seen in Figures 3a (MNIST) and 3b (CIFAR10). The first thing we notice is that for most ratios for the test SNR to training SNR we see sample wise double descent. Further, we see that the optimal denoising error does not occur when the train SNR is equal to the test SNR. This is very surprising as it contradicts standard thought that training data distribution should be the same as the test data distribution. Interestingly, as seen in Figures 4 and 4b, we see that the optimal ratio depends on the number of data points and the shape of the curve for the values $\hat{\theta}_{trn}/\hat{\theta}_{tst}$ that results in the best generalization error versus the number data points also has the shape of a double descent curve.

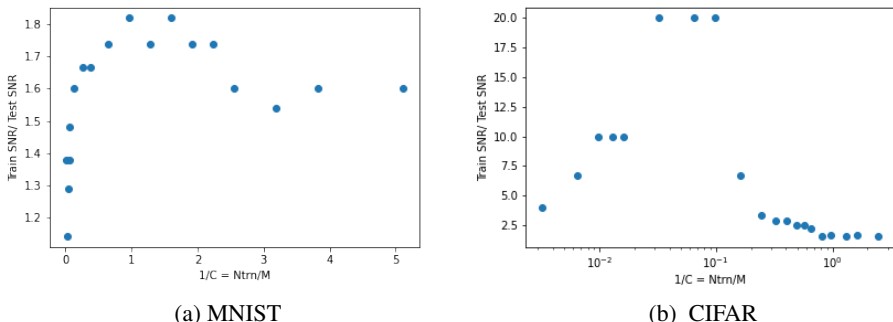

(a) MNIST  (b) CIFAR

Figure 4: Figure showing the sample wise double descent for the optimal amount of training noise.

## 4 THEORETICAL RESULTS AND CONSEQUENCES

In this paper, we want to theoretically understand the phenomena seen in Section 3. The main theoretical result of the paper is summarized below in Theorem 1.

**Theorem 1.** *Let $\sigma_i^{trn}, \sigma_i^{tst}$ be entries of $\Sigma_{trn}, \Sigma_{tst}$ and let $r = 1$. Let $c = M/N_{trn}$ be fixed. Suppose $\theta_{trn}$ is $O(\sqrt{N_{trn}})$ and $\theta_{tst}$ is $O(\sqrt{N_{tst}})$ Then, if $c < 1$, we have that*

$$R(\theta_{trn}, \theta_{tst}, c, \Sigma_{trn}, \Sigma_{tst}) = \frac{(\theta_{tst}\sigma_1^{tst})^2}{N_{tst}(1 + (\theta_{trn}\sigma_1^{trn})^2 c)^2} + \frac{c^2((\theta_{trn}\sigma_1^{trn})^2 + (\theta_{trn}\sigma_1^{trn})^4)}{M(1 + (\theta_{trn}\sigma_1^{trn})^2 c)^2(1 - c)} + o(1)$$

(3)

*and if $c > 1$, we have that*

$$R(\theta_{trn}, \theta_{tst}, c, \Sigma_{trn}, \Sigma_{tst}) = \frac{(\theta_{tst}\sigma_1^{tst})^2}{N_{tst}(1 + (\theta_{trn}\sigma_1^{trn})^2)^2} + \frac{c(\theta_{trn}\sigma_1^{trn})^2}{M(1 + (\theta_{trn}\sigma_1^{trn})^2)(c - 1)} + o(1).$$ (4)

*The $o(1)$ error term goes to 0 as $N_{trn}, M \to \infty$.*

We could imagine that the rank $r$ version is the same as the above but with a summation over the rank as shown in the equations below. But, we shall see in Section 5 that this turns out to only be an approximation. [1]

$$R(\theta_{trn}, \theta_{tst}, c, \Sigma_{trn}, \Sigma_{tst}) \approx \sum_{i=1}^{r} \frac{(\theta_{tst}\sigma_i^{tst})^2}{N_{tst}(1 + (\theta_{trn}\sigma_i^{trn})^2 c)^2} + \frac{c^2((\theta_{trn}\sigma_i^{trn})^2 + (\theta_{trn}\sigma_i^{trn})^4)}{M(1 + (\theta_{trn}\sigma_i^{trn})^2 c)^2(1 - c)} + o(1)$$

(5)

$$R(\theta_{trn}, \theta_{tst}, c, \Sigma_{trn}, \Sigma_{tst}) \approx \sum_{i=1}^{r} \frac{(\theta_{tst}\sigma_i^{tst})^2}{N_{tst}(1 + (\theta_{trn}\sigma_i^{trn})^2)^2} + \frac{c(\theta_{trn}\sigma_i^{trn})^2}{M(1 + (\theta_{trn}\sigma_i^{trn})^2)(c - 1)} + o(1).$$

(6)

### 4.1 OPTIMAL AMOUNT OF NOISE

First, if we ignore the error term, we can differentiate the formula to get the following formula for the optimal training SNR.

$$\frac{\theta_{opt-trn}^2}{N_{trn}} = \begin{cases} \max\left(0, \frac{\theta_{tst}^2}{N_{tst}}\left(1 - \frac{c}{2-c}\right) - \frac{c}{M(2-c)}\right) & c < 1 \\ \max\left(0, 2\frac{\theta_{tst}^2}{N_{tst}}(c - 1) - \frac{1}{N_{trn}}\right) & c > 1 \end{cases}$$

(7)

We already see the surprising result that the optimal training SNR and the test SNR are not equal. This is surprising, as traditional philosophy is that the training data should be drawn from the same distribution as the test data. Here instead we see that the optimal training distribution actually depends $c$. Further, the formulas in Equation 7 also describe a double descent curve for $\theta_{opt-trn}/N_{trn}$ versus $c$ curve as shown in Figure 5b.

---

[1]Derivation and exact assumptions for when the formulas are accurate in the appendix.

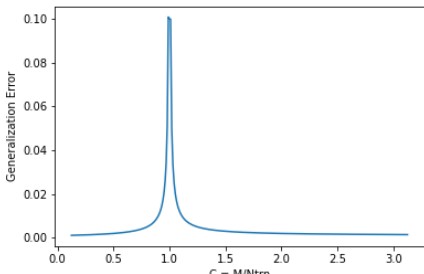
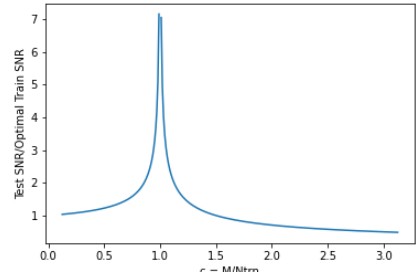

(a) Rank 1 Theory Generalization Error

(b) Rank 1 Theory Test SNR / Optimal training SNR

Figure 5: Plot showing the double descent curves for the generalization error as well as for the ratio of the test SNR to the optimal training SNR. Here $M = 1000$ and $\theta_{tst} = 1$ and $c$ was changed by changing $N_{trn}$.

## 4.2 DOUBLE DESCENT CURVES

We have already seen that the optimal amount of training noise follows a double decent curve. This is due to the double descent seen in the asymptotics for the generalization error. To understand this phenomenon, we first note that the bias of our model is given by the first term in formula in Theorem 1 and the variance is given by the second term. That is, we have that the variance given by

$$
\begin{cases}
\sum_{i=1}^{r} \frac{c^2((\theta_{trn}\sigma_i^{trn})^2 + (\theta_{trn}\sigma_i^{trn})^4)}{M(1+(\theta_{trn}\sigma_i^{trn})^2 c)^2(1-c)} & c < 1 \\
\sum_{i=1}^{r} \frac{c(\theta_{trn}\sigma_i^{trn})^2}{M(1+(\theta_{trn}\sigma_i^{trn})^2)(c-1)} & c > 1
\end{cases}
$$

From these formulas, we can see that as $c \to 1$ these formulas have a singularity. Since we have a linear model, $c = 1$ is the interpolation threshold (i.e., the point after which we have 0 training error). Hence as with previous models for double descent, we see that as we approach the interpolation threshold, the variance of model increases, resulting in an increase in the generalization error.

If we fix the number of features $M$ and change $c$ by varying $N_{trn}$, and also scale $\theta_{trn}$ as $\theta_{trn} = \hat{\theta}_{trn}\sqrt{N_{trn}}$, then we see that as $c \to 1$, the variance of the model increases. Once we have enough data points so that $c < 1$, we have the variance of the model starts decreasing. Additionally, we see that as we increase the number of data points, the bias decreases until we hit the interpolation threshold, after this point, the bias is constant. Similarly, if we fixed $N_{trn}$ and changed $c$ by changing $M$ then after the interpolation threshold, the inductive bias of the model kicks in. Here we see that the variance terms corresponds to $\|W\|_F^2$. Hence we see that as $c \to \infty$, we have that this implicitly regularize the weights of the network and get the second descent in the generalization error. That is, the variance of the model decreases as $c \to \infty$. Additionally, we see that as we increase the number of parameters, the bias of the model of the model decreases and then after the interpolation threshold it becomes constant. Note that this value is non-zero and depends on the training SNR.

In previous work (such as Mei & Montanari (2019)) on double descent curves for ridge regression, we see that optimal ridge regularization results in the vanishing of the double descent phenomena. This is also seen empirically for $L_2$ regularization for classification in Nakkiran et al. (2020). However, in our theoretical model even if we optimally pick the amount of training noise, we still have double descent. This is in contrast to results seen with a deep network on real data in Figure 3.

## 5 PROOF OF THEOREM 1

We prove Theorem 1, via the steps shown in Figure 6. The proofs for all of the lemmas have been moved to the appendix. Here we present a proof sketch that details the high-level ideas.

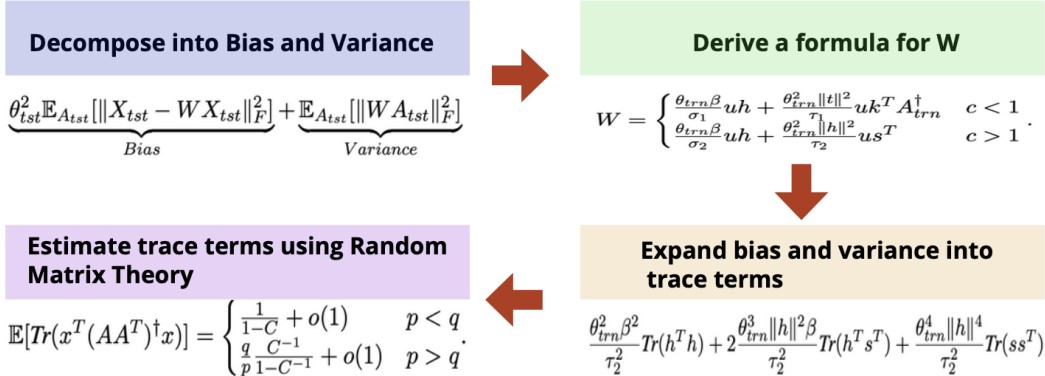

Figure 6: Figure showing the major steps used to derive the formula for the generalization error.

### 5.1 STEP 1: DECOMPOSE THE ERROR INTO BIAS AND VARIANCE TERMS

First, we decompose the error into its bias and variance.

**Lemma 1.** *If $A_{tst}$ has mean 0 entries and $A_{tst}$ is independent of $X_{tst}$ and $W$, then*

$$\mathbb{E}_{A_{tst}}[\|\theta_{tst} X_{tst} - W Y_{tst}\|_F^2] = \underbrace{\theta_{tst}^2 \mathbb{E}_{A_{tst}}[\|X_{tst} - W X_{tst}\|_F^2]}_{Bias} + \underbrace{\mathbb{E}_{A_{tst}}[\|W A_{tst}\|_F^2]}_{Variance}. \quad (8)$$

### 5.2 STEP 2: FORMULA FOR $W$

In our current setup, we know that $W$ is the solution to a least-squares problem. Hence $W = X_{trn} Y_{trn}^\dagger$. Expanding this out, we get the following formula for $W$. Let $h = v_{trn}^T A_{trn}^\dagger$, $k = A_{trn}^\dagger u$, $s = (I - A_{trn} A_{trn}^\dagger) u$, $t = v_{trn} (I - A_{trn}^\dagger A_{trn})$, $\beta = 1 + \theta_{trn} v_{trn}^T A_{trn}^\dagger u$, $\tau_1 = \theta_{trn}^2 \|t\|^2 \|k\|^2 + \beta^2$, and $\tau_2 = \theta_{trn}^2 \|s\|^2 \|h\|^2 + \beta^2$.

**Proposition 2.** *If $\beta \neq 0$ and $A_{trn}$ has full rank then*

$$W = \begin{cases} \frac{\theta_{trn}\beta}{\sigma_1} uh + \frac{\theta_{trn}^2 \|t\|^2}{\tau_1} uk^T A_{trn}^\dagger & c < 1 \\ \frac{\theta_{trn}\beta}{\sigma_2} uh + \frac{\theta_{trn}^2 \|h\|^2}{\tau_2} us^T & c > 1 \end{cases}.$$

For Gaussian noise $A_{trn}$ has full rank with probability 1 and $\beta$ is a random variable whose expected value is equal to 1, and the distribution is highly concentrated. Thus, Proposition 2 applies when $A_{trn}$ is isotropic Gaussian noise. Here we restricted ourselves to rank 1, as using Meyer (1973), we can expand formulas of the form $(A + xy^T)^\dagger$ where $x, y$ are vectors. For the higher rank case, we apply the formula form Meyer (1973) iteratively. This is the main difficulty of the method. Previous work on deriving asypmtotics for the generalization error had the noise on the output. Hence would take the pseudoinverse of a matrix that only depended on the data. However, in our case, we are taking the pseudoinverse of matrix that depends on the noise.

### 5.3 STEP 3:DECOMPOSE THE TERMS INTO SUM OF VARIOUS TRACE TERMS.

Let us first look at the bias term.

**Lemma 2.** *If $W$ is the solution to Equation 1, then*

$$X_{tst} - W X_{tst} = \begin{cases} \frac{\beta}{\tau_1} X_{tst} & \text{if } c < 1 \\ \frac{\beta}{\tau_2} X_{tst} & \text{if } c > 1 \end{cases}.$$

Let us now look at the second or the variance term.

**Lemma 3.** *If the entries of $A_{tst}$ are independent with mean 0, and variance $1/M$, then we have that* $\mathbb{E}_{A_{tst}}[\|W A_{tst}\|^2] = \frac{N_{tst}}{M} \|W\|^2$.

Note that this did not need any assumptions on $W$ or $X_{tst}$. All that was needed were the assumptions on $A_{tst}$. Thus, this holds more generally. This decomposition also follows from Bishop (1995). In light of Lemmas 1, 2, 3, and the fact that $\|X_{tst}\|_F^2 = \theta_{tst}^2$, we see that the expected mean squared generalization error is given by,

$$\mathbb{E}_{A_{tst}}\left[\frac{\|\theta_{tst}X_{tst} - WY_{tst}\|_F^2}{N_{tst}}\right] = \frac{1}{N_{tst}}\frac{\beta^2}{\tau_i^2}\theta_{tst}^2 + \frac{1}{M}\|W\|_F^2,$$

where $\tau_i$ depends on whether $c < 1$ or $c > 1$. Finally, let us look at the $\|W\|$ term.

**Lemma 4.** *If $\beta \neq 0$ and $A_{trn}$ has full rank, then we have that if $c < 1$,*

$$\|W\|_F^2 = \frac{\theta_{trn}^2\beta^2}{\tau_1^2}Tr(h^Th) + 2\frac{\theta_{trn}^3\|t\|^2\beta}{\tau_1^2}Tr(h^Tk^TA_{trn}^\dagger) + \frac{\theta_{trn}^4\|t\|^4}{\tau_1^2}Tr((A_{trn}^\dagger)^Tkk^TA_{trn}^\dagger)$$

*and if $c > 1$, then we have that*

$$\|W\|_F^2 = \frac{\theta_{trn}^2\beta^2}{\tau_2^2}Tr(h^Th) + 2\frac{\theta_{trn}^3\|h\|^2\beta}{\tau_2^2}Tr(h^Ts^T) + \frac{\theta_{trn}^4\|h\|^4}{\tau_2^2}Tr(ss^T).$$

### 5.4 STEP 4: ESTIMATE USING RANDOM MATRIX THEORY.

While the formula given by Lemmas 1, 3, and 4 is correct, we need a simpler formula to analyze the situation. Using ideas from random matrix theory, we can simplify the expression for $\|W\|_F^2$. To do so, we first need to prove Lemmas 5 and 6.

The main idea behind Lemmas 5 and 6 is that due to the rotational invariance of $A_{trn}$, the expectation of the trace of products of various matrices derived from $A_{trn}$ is determined by the expected value of some function $\chi$ of the eigenvalues of $A_{trn}$. However, instead of directly computing this expected value, we note that for any matrix $A$, that satisfies the noise assumptions, if we let $M, N \to \infty$, with $M/N \to c$, then the eigenvalue distribution converges to the Marchenko - Pastur distribution (Marcenko & Pastur, 1967; Götze & Tikhomirov, 2011; 2003; 2004; 2005; Bai et al., 2003). Götze & Tikhomirov (2004) showed that the distribution of the eigenvalues converged almost surely with a rate of at least $O(N^{-1/2+\epsilon})$ for any $\epsilon > 0$. Thus, we can use the expected value of the $\chi(\lambda)$ for $\lambda$ sampled from the Marchenko - Pastur distribution as an approximation.

For space reasons, we provide only one instance of the lemmas in the main text. The complete versions can be found in the appendix.

**Lemma 5.** *Suppose $A$ is an $p$ by $q$ matrix such that the entries of $A$ are independent and have mean 0, variance $1/q$, and bounded fourth moment. Let $W_p = AA^T$ and let $W_q = A^TA$. Let $C = p/q$. Suppose $\lambda_p, \lambda_q$ are a random eigenvalue of $W_p, W_q$. Then*

    *1. If $p < q$, then $\mathbb{E}\left[\frac{1}{\lambda_p}\right] = \frac{1}{1-C} + o(1)$.*

**Lemma 6.** *Suppose $A$ is an $p$ by $q$ matrix that satisfies the noise assumptions. Let $x, y$ be unit vectors in $p$ and $q$ dimensions. Let $C = p/q$. Then*

    *1. $\mathbb{E}[Tr(x^T(AA^T)^\dagger x)] = \begin{cases} \frac{1}{1-C} + o(1) & p < q \\ \frac{q}{p}\frac{C^{-1}}{1-C^{-1}} + o(1) & p > q \end{cases}$.*

Using these technical lemmas, we can now deal with all of the terms in the expressions in Lemma 4. First, let us look at the non-trace terms.

**Lemma 7.** *If $A_{trn}$ satisfies the noise assumptions, then we have that*

    *1. $\mathbb{E}[\beta/\theta_{trn}] = 1/\theta_{trn} + o(1)$ and $\mathrm{Var}(\beta/\theta_{trn}) = \frac{c}{(max(M,N_{trn})|1-c|)} + o(1)$.*

    *2. If $c < 1$, then $\mathbb{E}[\|h\|^2] = \frac{c^2}{1-c} + o(1)$ and $\mathrm{Var}(\|h\|^2) = \frac{c^3(2+c)}{N_{trn}(1-c)^3} + o(1)$.*

    *3. If $c > 1$, then $\mathbb{E}[\|h\|^2] = \frac{c}{c-1} + o(1)$ and $\mathrm{Var}(\|h\|^2) = \frac{c^2(2c-1)}{N_{trn}(c-1)^3} + o(1)$.*

    *4. $\mathbb{E}[\|k\|^2] = \frac{c}{1-c} + o(1)$ and $\mathrm{Var}(\|k\|^2) = \frac{c^2(2+c)}{M(1-c)^3} + o(1)$.*

5.  $\mathbb{E}[\|s\|^2] = \dfrac{c-1}{c} + o(1)$ *and* $\mathrm{Var}(\|s\|^2) = 2\dfrac{1}{Mc} + o(1)$

6.  $\mathbb{E}[\|t\|^2] = 1 - c + o(1)$, $\mathrm{Var}(\|t\|^2) = 2\dfrac{c}{N_{trn}} + o(1)$.

**Lemma 8.** *Under the noise assumptions, we have that*

$$\mathbb{E}[Tr(h^T k^T A_{trn}^\dagger)] = 0 \text{ and } \mathrm{Var}(Tr(h^T k^T A_{trn}^\dagger)) = \chi_3(c)/N_{trn},$$

*where* $\chi_3(c) = \mathbb{E}[1/\lambda^3]$, $\lambda$ *is an eigenvalue for* $AA^T$ *and* $A$ *is as in Lemma 6.*

**Lemma 9.** *Under the noise assumptions, we have that*

$$Tr((A_{trn}^\dagger)^T k k^T A_{trn}^\dagger) = \frac{c^2}{(1-c)^3} + o(1) \text{ and } \mathrm{Var}(Tr((A_{trn}^\dagger)^T k k^T A_{trn}^\dagger)) = \frac{3}{M}\chi_4(c) - \frac{1}{M}\frac{c^4}{(1-c)^6}$$

*where* $\chi_4(c) = \mathbb{E}[1/\lambda^4]$, $\lambda$ *is an eigenvalue for* $AA^T$ *and* $A$ *is as in Lemma 6.*

**Lemma 10.** *Under the same assumptions as Proposition 2, we have that* $Tr(h^T s^T) = 0$.

Lemmas 7, 8, 9, and 10 tell us that all of the terms are highly concentrated. Thus, even though such terms may not be uncorrelated, we can use the fact that $|\mathbb{E}[XY] - \mathbb{E}[X]\mathbb{E}[Y]| < \sqrt{\mathrm{Var}(X)\mathrm{Var}(Y)}$, to treat the terms as if they are uncorrelated. Since these variances have now been shown to be $o(1)$, we have that for each of these terms $\mathbb{E}[XY] = \mathbb{E}[X]\mathbb{E}[Y] + o(1)$. For example, since $\tau_1 = \beta^2 + \theta_{trn}^2\|t\|^2\|k\|^2 + o(1)$, using Lemmas 1, 4, and 6, we have that $\mathbb{E}[\tau_1] = 1 + \theta_{trn}^2 c + o(1)$. Similarly, $\mathbb{E}[\tau_2] = 1 + \theta_{trn}^2 + o(1)$. Finally, using these lemmas, we can simplify the expressions in Lemma 4 to get the formulas for the expected generalization error shown in Equations 3 and 4.

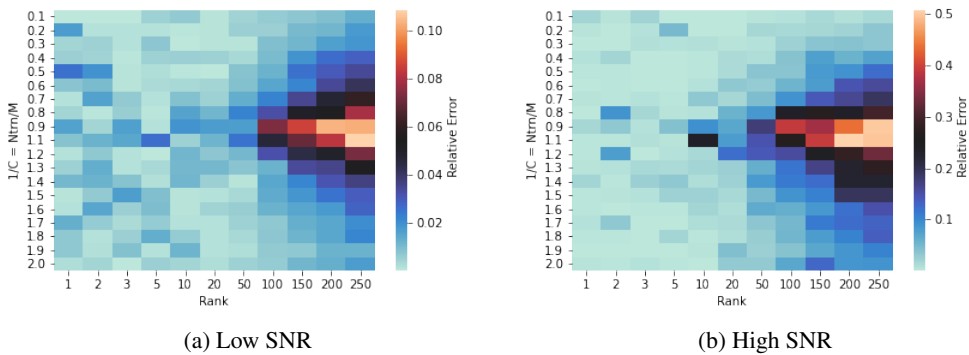

(a) Low SNR                                                          (b) High SNR

Figure 7: Figure showing the relative error for our formula. $M = 2500$ and $c$ is changed by changing $N_{trn}$. For low rank, we average over 10 trials and for high rank, we average over 100 trials.

## 6  ACCURACY OF APPROXIMATION

In this section we experimentally verify the accuracy of our formula for general rank $r$ data. Here for low SNR ($\theta_{trn}, \theta_{tst}$ are $O(1)$), we sample $\sigma_i^{trn}, \sigma_i^{tst}$ I.I.D. from the squared standard Gaussian and for high SNR ($\theta_{trn}, \theta_{tst}$ are $\Theta(\sqrt{N_{trn}}), \Theta(\sqrt{N_{tst}})$), we multiply this by $\sqrt{N_{trn}}, \sqrt{N_{tst}}$. As we can from Figure 7, we see that our formula is better for low SNR and low rank data.

## 7  CONCLUSION

In this paper, we switch focus from supervised set up to the unsupervised set up. Specifically, we look at the problem of denoising data. We empirically show that sample wise double descent exists for the generalization error. Further, we show that the optimal amount of training noise is not the same as the test noise. In fact, we see sample wise double descent for the ratio for the test SNR to the optimal training noise. To understand this phenomena, we study the simplest model, denoising rank 1 data using a linear model. Here we derive the exact asymptotics for the generalization error.

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

In this section we present all of the proofs for the results in the main text. Here we present the proofs in the same order they appear in the text.

## A   NOISE ASSUMPTIONS

**Proposition 1.** *If $B$ is a random matrix that has full rank with probability 1 and its entries are independent, have mean 0, and have variance $1/M$ and $P, Q$ are uniformly random orthogonal matrices. Then $A = PBQ$ satisfies all of our noise assumptions.*

*Proof.* Since $P, Q$ are a uniformly random orthogonal matrices, and $A = PBQ$, then it is clear that $A$ is rotationally bi-invariant and has full rank.

Since each entry of $B$ has mean 0 and each entry of $A$ is a linear combination of entries of $B$ where the coefficients (i.e., the entries from $P, Q$ are independent of $B$), we have that each entry of $B$ have mean 0. Due to the orthogonal nature of $P, Q$, we have the variance for an entry of $A$ is the same as the variance of entry in $B$.

Thus, the only thing left to prove is that the entries of $A$ are uncorrelated. To do this, we note that

$$a_{ij} = \sum_{k=1}^{N} \sum_{l=1}^{M} p_{il} b_{lk} q_{kj}.$$

Consider two entries $a_{i_1 j_1}$ and $a_{i_2 j_2}$. Then we have that

$$\mathbb{E}[a_{i_1 j_1} a_{i_2 j_2}] = \mathbb{E}\left[ \left( \sum_{k=1}^{N} \sum_{l=1}^{M} p_{i_1 l} b_{lk} q_{kj_1} \right) \left( \sum_{k=1}^{N} \sum_{l=1}^{M} p_{i_2 l} b_{lk} q_{kj_2} \right) \right]$$

$$= \sum_{k=1}^{N} \sum_{l=1}^{M} \mathbb{E}[p_{i_1 l} p_{i_2 l}] \mathbb{E}[b_{lk}^2] \mathbb{E}[q_{kj_1} q_{kj_2}]$$

$$= \frac{1}{M} \mathbb{E}\left[ \sum_{l=1}^{M} p_{i_1 l} p_{i_2 l} \right] \mathbb{E}\left[ \sum_{k=1}^{N} q_{kj_1} q_{kj_2} \right].$$

The second inequality follows from the fact that $P, Q, B$ are independent from each other, and that fact that the entries of $B$ are independent and have mean 0. Hence the cross terms have expectation 0. If we have that $i_1 = i_2$ and $j_1 \neq j_2$, then we have that since $Q$ is an orthogonal matrix

$$\sum_{k=1}^{N} \mathbb{E}[q_{kj_1} q_{kj_2}] = \mathbb{E}\left[ \sum_{k=1}^{N} q_{kj_1} q_{kj_2} \right] = 0.$$

Thus, the entries are uncorrelated. Similarly when $i_1 \neq i_2$ since $P$ is orthogonal matrix, we get that the entries are uncorrelated. □

**Convergence to Marchenko-Pastur.** If we strengthened the uncorrelated condition, to the entries being independent. Then due to the mean and variance assumptions (along with an assumption that the fourth moment is bounded), we would have convergence to Marchenko-Pastur distribution. However, the independence along with the bi-invariance would then force our noise distribution to be i.i.d. Gaussian.

In general however, with relaxed assumption of the entries only being uncorrelated, convergence is not known. However, in our case, we have a much simpler proof for matrices formed by Proposition 1. In our case, the noise matrices $B$ satisfy the standard assumptions for convergence. We then multiply $B$ by orthogonal matrices that are independent to $B$. Hence this has no effect on the eigenvalue distribution. Thus, the eigenvalues distribution for these matrices also converge to the Marchenko-Pastur distribution.

## B   PROOFS

Due to our data generation assumptions that $\|\Sigma_{trn}\|_F = \|\Sigma_{tst}\|_F = 1$ for rank 1 data, we have that $\sigma_1^{trn} = \sigma_1^{tst} = 1$.

## B.1 STEP ??: DECOMPOSE INTO BIAS AND VARAINCE

**Lemma 1.** *If $A_{tst}$ has mean 0 entries and $A_{tst}$ is independent of $X_{tst}$ and $W$, then*

$$\mathbb{E}_{A_{tst}}[\|X_{tst} - WY_{tst}\|_F^2] = \underbrace{\mathbb{E}_{A_{tst}}[\|X_{tst} - WX_{tst}\|_F^2]}_{Bias} + \underbrace{\mathbb{E}_{A_{tst}}[\|WA_{tst}\|_F^2]}_{Variance}.$$

*Proof.* Using the fact that for any two matrices $\|G - H\|_F^2 = \|G\|_F^2 + \|H\|_F^2 - 2\text{Tr}(G^T H)$, we get that

$$\|X_{tst} - WY_{tst}\|^2 = \|X_{tst} - WX_{tst} - WA_{tst}\|_F^2$$
$$= \|X_{tst} - WX_{tst}\|_F^2 + \|WA_{tst}\|^2 - 2\text{Tr}((X_{tst} - WX_{tst})^T WA_{tst}).$$

Then since the trace is linear, and $X_{tst}, W$ are independent of $A_{tst}$, and $A_{tst}$ has mean 0 entries, we see that

$$\mathbb{E}_{A_{tst}}[\text{Tr}((X_{tst} - WX_{tst})^T WA_{tst})] = 0.$$

Thus, we have the needed result. $\qquad\square$

## B.2 STEP ??: FORMULA FOR $W_{opt}$

**Proposition 2.** *Let $h = v_{trn}^T A_{trn}^\dagger$, $k = A_{trn}^\dagger u$, $s = (I - A_{trn}A_{trn}^\dagger)u$, $t = v_{trn}(I - A_{trn}^\dagger A_{trn})$, $\beta = 1 + \theta_{trn}v_{trn}^T A_{trn}^\dagger u$, $\tau_1 = \theta_{trn}^2\|t\|^2\|k\|^2 + \beta^2$, and $\tau_2 = \theta_{trn}^2\|s\|^2\|h\|^2 + \beta^2$. If $\beta \neq 0$ and $A_{trn}$ has full rank then*

$$W_{opt} = \begin{cases} \frac{\theta_{trn}\beta}{\tau_1}uh + \frac{\theta_{trn}^2\|t\|^2}{\tau_1}uk^T A_{trn}^\dagger & c < 1 \\ \frac{\theta_{trn}\beta}{\tau_2}uh + \frac{\theta_{trn}^2\|h\|^2}{\tau_2}us^T & c > 1 \end{cases}.$$

*Proof.* Let us first proof the case when $c > 1$. Here we know that $u$ is arbitrary. Here we have that $A_{trn}$ has full rank. Thus, since $c > 1$, we have that $M > N_{trn}$, thus $A_{trn}$ has rank $N_{trn}$. Thus, the rows of $A_{trn}$ span the whole space. Thus, $v_{trn}$ lives in the range of $A_{trn}^T$. Finally, since $\beta \neq 0$, we want Theorem 5 from Meyer (1973).

Here let us further define

$$p_2 = -\frac{\theta_{trn}^2\|s\|^2}{\beta}A_{trn}^\dagger h^T - \theta_{trn}k \text{ and } q_2^T = -\frac{\theta_{trn}\|h\|^2}{\beta}s^T - h$$

and finally $\tau_2 = \theta_{trn}^2\|s\|^2\|h\|^2 + \beta^2$. Then we have from Meyer (1973) that

$$(A_{trn} + \theta_{trn}uv_{trn}^T)^\dagger = A_{trn}^\dagger + \frac{\theta_{trn}}{\beta}A_{trn}^\dagger h^T s^T - \frac{\beta}{\tau_2}p_2 q_2^T$$

In our case, we only care about $\theta_{trn}uv_{trn}^T(A_{trn} + \theta_{trn}uv_{trn}^T)^\dagger$. Thus let us multiply this through and see what we get.

$$\theta_{trn}uv_{trn}^T(A_{trn} + \theta_{trn}uv_{trn}^T)^\dagger = \theta_{trn}uv_{trn}^T(A_{trn}^\dagger + \frac{\theta_{trn}}{\beta}A^\dagger h^T s^T - \frac{\beta}{\tau_2}p_2 q_2^T)$$

$$= \theta_{trn}uh + \frac{\theta_{trn}^2\|h\|^2}{\beta}us^T + \frac{\theta_{trn}\beta}{\tau_2}uv_{trn}^T\left(\frac{\theta_{trn}^2\|s\|^2}{\beta}A_{trn}^\dagger h^T + \theta_{trn}k\right)q_2^T$$

$$= \theta_{trn}uh + \frac{\theta_{trn}^2\|h\|^2}{\beta}us^T + \frac{\theta_{trn}^3\|s\|^2\|h\|^2}{\tau_2}uq_2^T + \frac{\theta_{trn}^2\beta}{\tau_2}uhuq_2^T$$

Then we have that

$$\frac{\theta_{trn}^3\|s\|^2\|h\|^2}{\tau_2}cq_2^T = -\frac{\theta_{trn}^4\|s\|^2\|h\|^4}{\tau_2\beta}us^T - \frac{\theta_{trn}^3\|s\|^2\|h\|^2}{\tau_2}uh \qquad (9)$$

and

$$\frac{\theta_{trn}^2 \beta}{\tau_2} uhuq_2^T = -\frac{\theta_{trn}^3 \|h\|^2}{\tau_2} uhus^T - \frac{\theta_{trn}^2 \beta}{\tau_2} uhuh. \tag{10}$$

Using that $\beta - 1 = \theta_{trn} v_{trn}^T A_{trn}^\dagger u = \theta_{trn} hu$, we get that

$$\frac{\theta_{trn}^2 \beta}{\tau_2} uhuq_2^T = -\frac{\theta_{trn}^2 \|h\|^2 (\beta-1)}{\tau_2} us^T - \frac{\theta_{trn} \beta (\beta-1)}{\tau_2} uh. \tag{11}$$

Substituting back in and collecting like terms we get that

$$\theta_{trn} uv_{trn}^T (A_{trn} + \theta_{trn} uv_{trn}^T)^\dagger = \theta_{trn} u \left( 1 - \frac{\theta_{trn}^2 \|s\|^2 \|h\|^2}{\tau_2} - \frac{\beta(\beta-1)}{\tau_2} \right) h +$$
$$\theta_{trn}^2 u \left( \frac{\|h\|^2}{\beta} - \frac{\theta_{trn}^2 \|s\|^2 \|h\|^4}{\tau_2 \beta} - \frac{\|h\|^2 (\beta-1)}{\tau_2} \right) s^T$$

We can then simplify the constants as follows.

$$1 - \frac{\theta_{trn}^2 \|s\|^2 \|h\|^2}{\tau_2} - \frac{\beta(\beta-1)}{\tau_2} = \frac{\tau_2 - \theta_{trn}^2 \|s\|^2 \|h\|^2 - \beta^2 + \beta}{\tau_2} = \frac{\beta}{\tau_2}$$

and

$$\frac{\|h\|^2}{\beta} - \frac{\theta_{trn}^2 \|s\|^2 \|h\|^4}{\tau_2 \beta} - \frac{\|h\|^2 (\beta-1)}{\tau_2} = \frac{\|h\|^2 (\tau_2 - \theta_{trn}^2 \|s\|^2 \|h\|^2 - \beta(\beta-1))}{\beta \tau_2} = \frac{\|h\|^2 \beta}{\beta \tau_2} = \frac{\|h\|^2}{\tau_2}.$$

This gives us the result for $c < 1$.

If $c > 1$, then we have that $M < N_{trn}$. Thus, the rank of $A_{trn}$ is $M$ the range of $A_{trn}$ is the whole space. Thus, $u$ lives in the range of $A_{trn}$. In this case, we then want Theorem 3 from Meyer (1973). In this case, we define

$$p_1 = -\frac{\theta_{trn}^2 \|k\|^2}{\beta} t^T - k \text{ and } q_1^T = -\frac{\theta_{trn} \|t\|^2}{\beta} k^T A_{trn}^\dagger - h.$$

Then in this case, we have that

$$(A_{trn} + \theta_{trn} uv_{trn}^T)^\dagger = A_{trn}^\dagger + \frac{\theta_{trn}}{\beta} t^T k^T A_{trn}^\dagger - \frac{\beta}{\tau_1} p_1 q_1^T.$$

Then we simplify the equation as we did before! □

## B.3 STEP ??: EXPAND INTO TRACE TERMS

**Lemma 3.** *If the entries of $A_{tst}$ are independent with mean 0, and variance $1/M$, then we have that* $\mathbb{E}_{A_{tst}}[\|WA_{tst}\|^2] = \frac{N_{tst}}{M} \|W\|^2$.

*Proof.* To see this, we note if we look at $A_{tst} A_{tst}^T$, then this is a $M$ by $M$, for which the expected value of the off diagonal entries is equal to 0, while the expected value of each diagonal entry is $N_{tst}/M$. That is, $\mathbb{E}_{A_{tst}}[A_{tst} A_{tst}^T] = \frac{N_{tst}}{M} I_M$.

Then note that

$$\|WA_{tst}\|^2 = \text{Tr}(A_{tst}^T W^T W A_{tst}) = \text{Tr}(W^T W A_{tst} A_{tst}^T) = \text{Tr}(W^T W A_{tst} A_{tst}^T).$$

Using the fact that the trace is linear again, we see that

$$\mathbb{E}_{A_{tst}}[\text{Tr}(W^T W A_{tst} A_{tst}^T)] = \text{Tr}(W^T W \mathbb{E}_{A_{tst}}[A_{tst} A_{tst}^T]) = \frac{N_{tst}}{M} \text{Tr}(W^T W) = \frac{N_{tst}}{M} \|W\|_F^2.$$

□

**Lemma 2.** *If $W$ is the solution to Equation 1, then*

$$X_{tst} - W X_{tst} = \begin{cases} \frac{\beta}{\tau_1} X_{tst} & \text{if } c < 1 \\ \frac{\beta}{\tau_2} X_{tst} & \text{if } c > 1 \end{cases}.$$

*Proof.* To see this, we have the following calculation for when $N_{trn} > M$.

$$X_{tst} - W X_{tst} = X_{tst} - \frac{\theta_{trn}\theta_{tst}\beta}{\tau_1} u h u v_{tst}^T - \frac{\theta_{trn}^2 \theta_{tst} \|t\|^2}{\tau_1} u k^T A_{trn}^\dagger u v_{tst}^T$$

$$= X_{tst} - \frac{\theta_{trn}\theta_{tst}\beta}{\tau_1} u v_{trn}^T A_{trn}^\dagger u v_{tst}^T - \frac{\theta_{trn}^2 \theta_{tst} \|t\|^2}{\tau_1} u k^T A_{trn}^\dagger u v_{tst}^T.$$

First, we note that $\beta = 1 + \theta_{trn} v_{trn}^T A_{trn}^\dagger u$. Thus, we have that $\theta v_{trn}^T A_{trn}^\dagger u = \beta - 1$. Thus, substituting this into the second term, we get that

$$X_{tst} - W X_{tst} = X_{tst} - \frac{\theta_{tst}\beta(\beta - 1)}{\tau_1} u v_{tst}^T - \frac{\theta_{trn}^2 \theta_{tst} \|t\|^2}{\tau_1} u k^T A_{trn}^\dagger u v_{tst}^T.$$

For the third term, we note that $k = A_{trn}^\dagger u$. Thus, we have that $k^T A_{trn}^\dagger u = k^T k = \|k\|^2$. Substituting this into the expression, we get that

$$X_{tst} - W X_{tst} = X_{tst} - \frac{\theta_{tst}\beta(\beta - 1)}{\tau_1} u v_{tst}^T - \frac{\theta_{trn}^2 \theta_{tst} \|t\|^2 \|k\|^2}{\tau_1} u v_{tst}^T.$$

Noting that $X_{tst} = \theta_{tst} u v_{tst}^T$, we get that

$$X_{tst} - W X_{tst} = X_{tst} \left( 1 - \frac{\beta(\beta - 1)}{\tau_1} - \frac{\theta_{trn}^2 \|t\|^2 \|k\|^2}{\tau_1} \right).$$

To simplify the constants, we note that $\tau_1 = \theta_{trn}^2 \|t\|^2 \|k\|^2 + \beta^2$. Thus, we get that

$$\frac{\tau_1 + \beta - \beta^2 - \theta_{trn}^2 \|t\|^2 \|k\|^2}{\tau_1} = \frac{\beta}{\tau_1}.$$

For the case when $N_{trn} < M$, we note that the first term of $W$ is the same (modulo replacing $\tau_1$ for $\tau_2$) as it is for the case when $c > 1$. Thus, we just need to deal with the last term. Here we see that the last term is

$$\frac{\theta_{trn}^2 \theta_{tst} \|h\|^2}{\tau_2} u s^T u v_{tst}^T.$$

Here we note that $s = (I - A_{trn} A_{trn}^\dagger) u$. Thus, in particular, $s$ is the projection of $u$ onto the kernel of $A_{trn}^T$. Thus, we have that $u = s + \hat{s}$, where $s \perp \hat{s}$. This then tells us that $s^T u = \|s\|^2$. Thus, for this term, we get that it is equal to

$$\frac{\theta^2 \|h\|^2 \|s\|^2}{\tau_2} X_{tst}.$$

For this term we note that $\tau_2 = \beta^2 + \theta^2 \|h\|^2 \|u\|^2$. Thus, doing the same simplification as before, we see that for the case when $N_{trn} < M$, we have that

$$X_{tst} - W X_{tst} = \frac{\beta}{\tau_2} X_{tst}.$$

$\square$

In light of Lemma 2 and the fact that $\|X_{tst}\|_F^2 = \theta_{tst}^2$. We see that if we look at the expected MSE, we have that,

$$\mathbb{E}_{A_{tst}} \left[ \frac{\|X_{tst} - W(X_{tst} + A_{tst})\|}{N_{tst}} \right] = \frac{\beta}{N_{tst}\tau_i} \theta_{tst}^2 + \frac{1}{M} \|W\|_F^2,$$

where $\tau_i$ depends on whether $c < 1$ or $c > 1$.

Finally, let us look at the $\|W\|$ term.

**Lemma 4.** *If $\beta \neq 0$ and $A_{trn}$ has full rank, then we have that if $c < 1$,*

$$\|W\|_F^2 = \frac{\theta_{trn}^2 \beta^2}{\tau_1^2} Tr(h^T h) + 2\frac{\theta_{trn}^3 \|t\|^2 \beta}{\tau_1^2} Tr(h^T k^T A_{trn}^\dagger) + \frac{\theta_{trn}^4 \|t\|^4}{\tau_1^2} Tr((A_{trn}^\dagger)^T k k^T A_{trn}^\dagger)$$

*and if $c > 1$, then we have that*

$$\|W\|_F^2 = \frac{\theta_{trn}^2 \beta^2}{\tau_2^2} Tr(h^T h) + 2\frac{\theta_{trn}^3 \|h\|^2 \beta}{\tau_2^2} Tr(h^T s^T) + \frac{\theta_{trn}^4 \|h\|^4}{\tau_2^2} Tr(ss^T).$$

*Proof.* To deal with the term $\text{Tr}(W^T W)$ we are again going to have to look at whether $N_{trn}$ is bigger than or smaller than $M$. First, let us start by looking at the case when $N_{trn} > M$. Here we have that

$$\|W\|_F^2 = \text{Tr}(W^T W)$$

$$= \text{Tr}\left(\left(\frac{\theta_{trn}\beta}{\tau_1}uh + \frac{\theta_{trn}^2 \|t\|^2}{\tau_1}uk^T A_{trn}^\dagger\right)^T \left(\frac{\theta_{trn}\beta}{\tau_1}uh + \frac{\theta_{trn}^2 \|t\|^2}{\tau_1}uk^T A_{trn}^\dagger\right)\right)$$

$$= \frac{\theta_{trn}^2 \beta^2}{\tau_1^2}\text{Tr}(h^T u^T uh) + 2\frac{\theta_{trn}^3 \|t\|^2 \beta}{\tau_1^2}\text{Tr}(h^T u^T u k^T A_{trn}^\dagger) + \frac{\theta_{trn}^4 \|t\|^4}{\tau_1^2}\text{Tr}((A_{trn}^\dagger)^T k u^T u k^T A_{trn}^\dagger)$$

$$= \frac{\theta_{trn}^2 \beta^2}{\tau_1^2}\text{Tr}(h^T h) + 2\frac{\theta_{trn}^3 \|t\|^2 \beta}{\tau_1^2}\text{Tr}(h^T k^T A_{trn}^\dagger) + \frac{\theta_{trn}^4 \|t\|^4}{\tau_1^2}\text{Tr}((A_{trn}^\dagger)^T k k^T A_{trn}^\dagger).$$

Where the last inequality is true due to the fact that $\|u\|^2 = 1$. How about when $N_{trn} < M$. Then we have the following string of equalities instead.

$$\|W\|_F^2 = \text{Tr}(W^T W)$$

$$= \text{Tr}\left(\left(\frac{\theta_{trn}\beta}{\tau_2}uh + \frac{\theta_{trn}^2 \|h\|^2}{\tau_2}us^T\right)^T \left(\frac{\theta_{trn}\beta}{\tau_2}uh + \frac{\theta_{trn}^2 \|h\|^2}{\tau_2}us^T\right)\right)$$

$$= \frac{\theta_{trn}^2 \beta^2}{\tau_2^2}\text{Tr}(h^T u^T uh) + 2\frac{\theta_{trn}^3 \|h\|^2 \beta}{\tau_2^2}\text{Tr}(h^T u^T us^T) + \frac{\theta_{trn}^4 \|h\|^4}{\tau_1^2}\text{Tr}(su^T us^T)$$

$$= \frac{\theta_{trn}^2 \beta^2}{\tau_2^2}\text{Tr}(h^T h) + 2\frac{\theta_{trn}^3 \|h\|^2 \beta}{\tau_2^2}\text{Tr}(h^T s^T) + \frac{\theta_{trn}^4 \|h\|^4}{\tau_2^2}\text{Tr}(ss^T).$$

$\square$

### B.4 STEP ??: ESTIMATE USING RANDOM MATRIX THEORY.

**Lemma 5.** *Suppose $A$ is an $p$ by $q$ matrix such that the entries of $A$ are independent and have mean 0, variance $1/q$, and bounded fourth moment. Let $W_p = AA^T$ and let $W_q = A^T A$. Let $C = p/q$. Suppose $\lambda_p, \lambda_q$ are a random eigenvalue of $W_p, W_q$. Then*

*1. If $p < q$, then $\mathbb{E}\left[\frac{1}{\lambda_p}\right] = \frac{1}{1-C} + o(1)$.*

*2. If $p < q$, then $\mathbb{E}\left[\frac{1}{\lambda_p^2}\right] = \frac{1}{(1-C)^3} + o(1)$.*

*3. If $p < q$, then $\mathbb{E}\left[\frac{1}{\lambda_p^3}\right] = \frac{1}{(1-C)^5} + o(1)$.*

*4. If $p < q$, then $\mathbb{E}\left[\frac{1}{\lambda_p^4}\right] = \frac{C^2 + \frac{22}{6}c + 1}{(1-C)^7} + o(1)$.*

*5. If $p > q$, then $\mathbb{E}\left[\frac{1}{\lambda_q}\right] = \frac{C^{-1}}{1-C^{-1}} + o(1)$.*

*6. If $p > q$, then $\mathbb{E}\left[\frac{1}{\lambda_q^2}\right] = \frac{C^{-2}}{(1-C^{-1})^3} + o(1)$.*

7. If $p > q$, then $\mathbb{E}\left[\frac{1}{\lambda_p^3}\right] = \frac{C^{-3}(1+C^{-1})}{(1-C^{-1})^5} + o(1)$.

8. If $p > q$, then $\mathbb{E}\left[\frac{1}{\lambda_p^4}\right] = \frac{C^{-4}(C^{-2}+\frac{22}{6}C^{-1}+1)}{(1-C^{-1})^7} + o(1)$.

*Proof.* Suppose $A$ is an $p$ by $q$ matrix such that the entries of $A$ are independent and have mean 0, variance $1/q$, and bounded fourth moment. Then we know that $W_p = AA^T$ is an $p$ by $p$ Wishart matrix with $c = C$. If we send $p, q$ to infinity such that $p/q$ remains constant, then we have the eigenvalue distribution $F_p$ converges to the Marchenko Pastur distribution $F$ in probability.

From Rao & Edelman (2008), we know there exists a bi variate polynomial $L(m, z) = czm^2 - (1 - c - z)m + 1$ such that the zeros of $L(m, z)$ given by $L(m(z), z)$ are such that

$$m(z) = \int \frac{1}{\lambda - z} dF(\lambda) = \mathbb{E}_\lambda\left[\frac{1}{\lambda - z}\right].$$

For the Marchenko-Pastur distribution, we have that for $z = 0$, we get that $m(z) = 1/(1 - c)$. Thus, for $\lambda_p$ is an eigenvalue value of $W_p$, we have that

$$\mathbb{E}\left[\frac{1}{\lambda_p}\right] = \frac{1}{1 - c} + o(1).$$

For $\mathbb{E}_\lambda\left[\frac{1}{(\lambda-z)^2}\right]$ we need to calculate $m'(0)$. Using the implicit function theorem, we know that

$$m'(z) = -1\left(\frac{\partial L}{\partial m}(m(z), z)\right)^{-1}\frac{\partial L}{\partial z}(m(z), z).$$

Here we can see that $\partial L/\partial m = 2czm + c + z - 1$. Thus, at $(1/(1 - c), 0)$, this is equal to $c - 1$. Also $\partial L/\partial z = cm^2 + m$. Again at $(1/(1 - c), 0)$ this is equal to $\frac{c}{(1-c)^2} + \frac{1}{1-c} = \frac{1}{(1-c)^2}$. Thus, we have that

$$m'(0) = \frac{1}{(1 - c)^3}.$$

Similarly, using the implicit function formulation, we can calculate $m''(0)$ and $m'''(0)$.

On the other hand if $q < p$, then $W_q := A^T A$ is not a Wishart matrix here, because it is scaled by the wrong constant. However, multiplying it by $1/C$ gives us the correct scaling. Thus, $A^T A/C$ is a Wishart matrix with $c = 1/C$ Thus, for $\lambda_q$ is an eigenvalue value of $W_q$, we have that

$$\mathbb{E}\left[\frac{1}{\lambda_q}\right] = \frac{C^{-1}}{1 - C^{-1}} + o(1).$$

We can obtain the rest in a similar manner from the previous results. $\qquad\square$

**Lemma 6.** *Suppose $A$ is an $p$ by $q$ matrix that satsifies the standard noise assumptions. Let $x, y$ be unit vectors in $p$ and $q$ dimensions. Let $C = p/q$. Then*

1. $\mathbb{E}[Tr(x^T(AA^T)^\dagger x)] = \begin{cases} \frac{1}{1-C} + o(1) & p < q \\ \frac{q}{p}\frac{C^{-1}}{1-C^{-1}} + o(1) & p > q \end{cases}$.

2. $\mathbb{E}[Tr(x^T(AA^T)^\dagger(AA^T)^\dagger x)] = \begin{cases} \frac{1}{(1-C)^3} + o(1) & p < q \\ \frac{q}{p}\frac{C^{-2}}{(1-C^{-1})^3} + o(1) & p > q \end{cases}$.

3. $\mathbb{E}[Tr(y^T(A^T A)^\dagger y)] = \begin{cases} \frac{p}{q}\frac{1}{1-C} + o(1) & p < q \\ \frac{C^{-1}}{1-C^{-1}} + o(1) & p > q \end{cases}$.

4. $\mathbb{E}[Tr(y^T(A^T A)^\dagger(A^T A)^\dagger y)] = \begin{cases} \frac{p}{q}\frac{1}{(1-C)^3} + o(1) & p < q \\ \frac{C^{-2}}{(1-C^{-1})^3} + o(1) & p > q \end{cases}$.

*Proof.* Let $A = U\Sigma V^T$ be the SVD. Then we have that $(AA^T)^\dagger = U(\Sigma^2)^\dagger U^T$. Then since $A$ is bi-unitary invariant, we have that $U$ is a uniformly random unitary matrix. Thus, $a = x^T U$ is a uniformly random unit vector. Note with probability 1, the rank of $A$ is full and that the non-zero eigenvalues of $A^T A$ and $AA^T$ are the same.

If $p < q$, then we have that

$$\mathbb{E}[\mathrm{Tr}(x^T(AA^T)^\dagger x)] = \sum_{i=1}^{p} a_i^2 \frac{1}{\sigma_i^2}.$$

Using Lemma 5, we have that $\mathbb{E}[1/_i^2] = 1/(1-C) + o(1)$. Thus, we have that

$$\mathbb{E}[\mathrm{Tr}(x^T(AA^T)^\dagger x)] = \sum_{i=1}^{p} \frac{1}{p}\frac{1}{1-C} + o(1).$$

On the other hand, if $p > q$, from Lemma 5, we have that $\mathbb{E}[1/_i^2] = C^{-1}/(1 - C^{-1}) + o(1)$. Thus,

$$\mathbb{E}[\mathrm{Tr}(x^T(AA^T)^\dagger x)] = \sum_{i=1}^{q} \frac{1}{p}\frac{C^{-1}}{1-C^{-1}} + o(1).$$

Similarly, if we had we looking at $\mathrm{Tr}(x^T(AA^T)^\dagger(AA^T)^\dagger x)$, we would have a $1/_i^4$ term instead. Thus, if $p < q$, we would have that

$$\mathbb{E}[\mathrm{Tr}(x^T(AA^T)^\dagger(AA^T)^\dagger x)] = \frac{1}{(1-C)^3} + o(1).$$

A similar calculation holds for the others. □

Now we have the following Lemma in the main text. However, here instead of having one big proof, we will separate each term out into its own lemma.

**Lemma 7.** *If $A_{trn}$ satisfies the standard noise assumptions, then we have that*

1. $\mathbb{E}[\beta] = 1 + o(1)$ *and* $\mathrm{Var}(\beta) = \frac{\theta_{trn}^2 c}{(max(M,N_{trn})|1-c|))} + o(1)$.
2. *If $c < 1$, then* $\mathbb{E}[\|h\|^2] = \frac{c^2}{1-c} + o(1)$ *and* $\mathrm{Var}(\|h\|^2) = \frac{c^3(2+c)}{N_{trn}(1-c)^3} + o(1)$.
3. *If $c > 1$, then* $\mathbb{E}[\|h\|^2] = \frac{c}{c-1} + o(1)$ *and* $\mathrm{Var}(\|h\|^2) = \frac{c^2(2c-1)}{N_{trn}(c-1)^3} + o(1)$.
4. $\mathbb{E}[\|k\|^2] = \frac{c}{1-c} + o(1)$ *and* $\mathrm{Var}(\|k\|^2) = \frac{c^2(2+c)}{M(1-c)^3} + o(1)$.
5. $\mathbb{E}[\|s\|^2] = \frac{c-1}{c} + o(1)$ *and* $\mathrm{Var}(\|s\|^2) = 2\frac{1}{Mc} + o(1)$
6. $\mathbb{E}[\|t\|^2] = 1 - c + o(1)$, $\mathrm{Var}(\|t\|^2) = 2\frac{c}{N_{trn}} + o(1)$.

**Lemma 11.** $\beta$ *term.*

*Proof.* First, we calculate the expected value of $\beta$. To do so, let $A_{trn} = U\Sigma V^T$ be the SVD. Then since $A_{trn}$ is bi-unitarily invariant, we have that $U, V$ are uniformly random unitary matrices. Since $u, v_{trn}$ are fixed. We have that $a := v_{trn}^T V \in \mathbb{R}^{N_{trn}}$ and $b := U^T u \in \mathbb{R}^M$ are uniformly random unit vectors. In particular, we have that $\mathbb{E}[a_i] = 0, \mathbb{E}[b_i] = 0, \mathrm{Var}(a_i) = 1/N_{trn}, \mathrm{Var}(b_i) = 1/M$.

Thus, if $\sigma_i$ are the singular values for $A_{trn}$, then we have that

$$\beta = 1 + \theta_{trn} \sum_{i=1}^{\min(M,N_{trn})} \frac{1}{\sigma_i} a_i b_i.$$

Thus, if you take the expectation you get that

$$\mathbb{E}[\beta] = 1.$$

On the other hand, lets look at the variance. For the variance, we need to compute $\mathbb{E}[\beta^2]$. Now if we let $T := \theta_{trn} v_{trn}^T A_{trn}^\dagger u$. Then we have that

$$\beta^2 = 1 + T^2 + 2T.$$

Thus, again if we take the expectation, we get that

$$\mathbb{E}[\beta^2] = 1 + \mathbb{E}[T^2].$$

Again due to the fact that $a, b$ are independent have have mean 0 entries, the cross terms in $\mathbb{E}[T^2]$. Thus, we have that

$$\mathbb{E}[T^2] = \theta trn^2 \mathbb{E}\left[ \sum_{i=1}^{\min(M,N_{trn})} \frac{1}{\sigma_i^2} a_i^2 b_i^2 \right] = \theta trn^2 \frac{1}{MN_{trn}} \mathbb{E}\left[ \sum_{i=1}^{\min(M,N_{trn})} \frac{1}{\sigma_i^2} \right].$$

Now we need to case on whether $M > N_{trn}$ or $M < N_{trn}$. Now to use Lemma 5, we note that $q = M$ and $p = N_{trn}$.

Suppose we have that $M > N_{trn}$, then in this case, we have that $q > p$. Thus, we have that

$$\mathbb{E}\left[\frac{1}{\sigma_i^2}\right] = \frac{1}{1-C} + o(1),$$

where $C = p/q = N_{trn}/M = 1/c$. Thus, we have that

$$\mathbb{E}\left[\frac{1}{\sigma_i^2}\right] = \frac{1}{1-1/c} + o(1) = \frac{c}{c-1} + o(1).$$

Thus, we have that

$$\mathbb{E}[T^2] = \theta_{trn}^2 \frac{c}{M(c-1)} + o\left(\frac{1}{M}\right).$$

Thus, we have

$$\mathrm{Var}(\beta) = \theta_{trn}^2 \frac{c}{M(c-1)} + o\left(\frac{1}{M}\right).$$

On the other hand, if $M < N_{trn}$. Then we have that $q < p$. Thus, we have that

$$\mathbb{E}\left[\frac{1}{\sigma_i^2}\right] = \frac{C^{-1}}{1-C^{-1}} + o(1),$$

where $C = p/q = N_{trn}/M = 1/c$. Thus, we have that

$$\mathbb{E}\left[\frac{1}{\sigma_i^2}\right] = \frac{c}{1-c} + o(1).$$

Thus, we have that

$$\mathbb{E}[T^2] = \theta_{trn}^2 \frac{1}{N_{trn}} \left( \frac{c}{1-c} + o(1) \right) = \frac{c}{N_{trn}(1-c)} + o\left(\frac{1}{N_{trn}}\right).$$

Thus, we have

$$\mathrm{Var}(\beta) = \theta_{trn}^2 \frac{c}{N_{trn}(1-c)} + o\left(\frac{1}{N_{trn}}\right).$$

$\square$

**Lemma 12.** $\|h\|^2$ *term.*

*Proof.* We want to do a calculation similar to that in Lemma 1. Here we have that

$$\|h\|^2 = \mathrm{Tr}(h^T h) = \mathrm{Tr}((A_{trn}^\dagger)^T v_{trn} v_{trn}^T A_{trn}^\dagger) = \mathrm{Tr}(v_{trn}^T A_{trn}^\dagger (A_{trn}^\dagger)^T v_{trn}) = \mathrm{Tr}(v_{trn}^T (A_{trn}^T A_{trn})^\dagger v_{trn}).$$

To use Lemma 6, we note that $A = A_{trn}^T$, $q = M$, $p = N_{trn}$. Let us now suppose that $M < N_{trn}$. Then again taking the expectation, we see that

$$\mathbb{E}[\|h\|^2] = \frac{M}{N_{trn}}\left(\frac{c}{1-c} + o(1)\right) = \frac{c^2}{1-c} + o(1).$$

For the expectation of $\|h\|^4$, let $A_{trn} = U\Sigma V^T$ be the svd. Then $h = v_{trn}^T V \Sigma^\dagger U^T$. Let $a = v_{trn}^T V$ and note that $a$ is a uniformly random unit vector. Thus, we have that

$$\|h\|^2 = \sum_{i=1}^{M} \frac{1}{\sigma_i^2} a_i^2.$$

For the expectation of $\|h\|^4$, we note that

$$\|h\|^4 = \sum_{i=1}^{M}\sum_{j=1}^{M} \frac{1}{\sigma_i^2 \sigma_j^2} a_i^2 a_j^2 = \sum_{i=1}^{M} \frac{1}{\sigma_i^4} a_i^4 + \sum_{i \neq j} \frac{1}{\sigma_i^2}\frac{1}{\sigma_j^2} a_i^2 a_j^2.$$

Taking the expectation of the first term, we get

$$\sum_{i=1}^{M} \mathbb{E}\left[\frac{1}{\sigma_i^4}\right]\mathbb{E}[a_i^4] = \frac{3M}{N_{trn}^2}\left(\frac{c^2}{(1-c)^3} + o(1)\right) = 3\frac{c^3}{N_{trn}(1-c)^3} + o(1).$$

Taking the expectation of the second term, we get

$$M(M-1)\mathbb{E}\left[\frac{1}{\sigma_i^2}\right]^2\mathbb{E}[a_i^2]^2 = M(M-1)\frac{1}{N_{trn}^2}\left(\frac{c^2}{(1-c)^2} + o(1)\right) = \frac{c^4}{(1-c)^2} - \frac{c^3}{N_{trn}(1-c)^2} + o(1).$$

Thus, we have that

$$\mathbb{E}[\|h\|^4] = \frac{c^4}{(1-c)^2} + \frac{c^3(2+c)}{N_{trn}(1-c)^3} + o(1).$$

Thus, the variance is

$$\mathrm{Var}(\|h\|^2) = \frac{c^3(2+c)}{N_{trn}(1-c)^3} + o(1).$$

For $M > N_{trn}$, we instead have that

$$\mathbb{E}[\|h\|^2] = \frac{N_{trn}}{N_{trn}}\left(\frac{c}{c-1} + o(1)\right) = \frac{c}{c-1} + o(1).$$

For the expectation of $\|h\|^4$, we note that

$$\|h\|^4 = \sum_{i=1}^{N_{trn}}\sum_{j=1}^{N_{trn}} \frac{1}{\sigma_i^2 \sigma_j^2} a_i^2 a_j^2 = \sum_{i=1}^{N_{trn}} \frac{1}{\sigma_i^4} a_i^4 + \sum_{i \neq j} \frac{1}{\sigma_i^2}\frac{1}{\sigma_j^2} a_i^2 a_j^2.$$

Taking the expectation of the first term, we get

$$\sum_{i=1}^{N_{trn}} \mathbb{E}\left[\frac{1}{\sigma_i^4}\right]\mathbb{E}[a_i^4] = \frac{3N_{trn}}{N_{trn}^2}\left(\frac{c^3}{(c-1)^3} + o(1)\right) = 3\frac{c^3}{N_{trn}(c-1)^3} + o(1).$$

Taking the expectation of the second term, we get

$$N_{trn}(N_{trn}-1)\mathbb{E}\left[\frac{1}{\sigma_i^2}\right]^2\mathbb{E}[a_i^2]^2 = N_{trn}(N_{trn}-1)\frac{1}{N_{trn}^2}\left(\frac{c^2}{(c-1)^2} + o(1)\right)$$

$$= \frac{c^2}{(c-1)^2} - \frac{c^2}{N_{trn}(c-1)^2} + o(1).$$

Thus, we have that

$$\mathbb{E}[\|h\|^4] = \frac{c^2}{(c-1)^2} + 3\frac{c^3}{N_{trn}(c-1)^3} - \frac{c^2}{N_{trn}(c-1)^2} + o(1) = \frac{c^2}{(c-1)^2} + \frac{c^2(2c-1)}{N_{trn}(c-1)^3} + o(1).$$

Thus, the variance is

$$\mathrm{Var}(\|h\|^2) = \frac{c^2(2c-1)}{N_{trn}(c-1)^3} + o(1).$$

$\square$

**Lemma 13.** $\|k\|^2$ *term.*

*Proof.* First note that $k$ only appears in the formula when $c < 1$. Thus, we can focus on this case. As with $h$, we have that

$$\|k\|^2 = \text{Tr}(u^T (A_{trn}^\dagger)^T A_{trn}^\dagger u) = \text{Tr}(u^T (A_{trn} A_{trn}^T)^\dagger u).$$

Again using Lemma 6, with $q = M, p = N_{trn}, A = A_{trn}, y = u$. Thus, since we have $q = M < N_{trn} = p$, we get that

$$\mathbb{E}[\|k\|^2] = \frac{c}{1-c} + o(1).$$

To calculate the variance, we need to calculate the expectation of $\|k\|^4$. Here be again let $A = U\Sigma V^T$ be the SVD. Then let $b := U^T u$. Then we have that

$$\|k\|^2 = \sum_{i=1}^M \frac{1}{\sigma_i^2} b_i^2.$$

Thus, we see that

$$\|k\|^4 = \sum_{i=1}^M \frac{1}{\sigma_i^4} b_i^4 + \sum_{i \neq j} \frac{1}{\sigma_i^2} \frac{1}{\sigma_j^2} b_i^2 b_j^2.$$

Taking the expectation of the first term we get

$$3 \frac{M}{M^2} \frac{c^2}{(1-c)^3} = \frac{3c^2}{M(1-c)^3}.$$

Taking the expectation of the second term we get

$$\frac{M(M-1)}{M^2} \frac{c^2}{(1-c)^2} = \frac{c^2}{(1-c)^2} - \frac{c^2}{M(1-c)^2}.$$

Thus, we have that

$$\mathbb{E}[\|k\|^4] = \frac{c^2}{(1-c)^2} + \frac{c^2(2+c)}{M(1-c)^3} + o(1).$$

Thus, we have that

$$\text{Var}(\|k\|^2) = \frac{c^2(2+c)}{M(1-c)^3} + o(1).$$

$\square$

**Lemma 14.** $\|s\|^2$ *term.*

*Proof.* First, we note that $s$ only appears when $M > N_{trn}$. Thus, we only need to deal with that case. For this term, we note that $(I - A_{trn} A_{trn}^\dagger)$ is a projection matrix onto a uniformly random $M - N_{trn}$ dimensional subspace. Here be again let $A = U\Sigma V^T$ be the SVD. Then let $b := U^T u$.

$$\mathbb{E}[\|s\|^2] = \mathbb{E}[u^T u - u^T A_{trn} A_{trn}^\dagger u] = \mathbb{E}\left[1 - b^T \begin{bmatrix} I_{N_{trn}} & 0 \\ 0 & 0 \end{bmatrix} b\right] = 1 - \sum_{i=1}^{N_{trn}} \frac{1}{M} = 1 - \frac{1}{c}$$

Similarly, we have that

$$\|s\|^4 = \left(1 - \sum_{i=1}^{N_{trn}} b_i^2\right)^2$$

$$= 1 + \left(\sum_{i=1}^{N_{trn}} b_i^2\right)^2 - 2 \sum_{i=1}^{N_{trn}} b_i^2$$

$$= 1 + \sum_{i=1}^{N_{trn}} b_i^4 + \sum_{i \neq j}^{N_{trn}} b_i^2 b_j^2 - 2 \sum_{i=1}^{N_{trn}} b_i^2$$

Taking the expectation, we get that

$$
\begin{aligned}
\mathbb{E}[\|s\|^4] &= 1 + 3 \sum_{i=1}^{N_{trn}} \frac{1}{M^2} + \sum_{i \neq j}^{N_{trn}} \frac{1}{M^2} - 2 \sum_{i=1}^{N_{trn}} \frac{1}{M} \\
&= 1 + \frac{3}{cM} + \frac{N_{trn}(N_{trn}-1)}{M^2} - 2\frac{1}{c} \\
&= 1 + \frac{3}{cM} + \frac{1}{c^2} - \frac{1}{cM} - 2\frac{1}{c} \\
&= \left(1 - \frac{1}{c}\right)^2 + \frac{2}{cM}
\end{aligned}
$$

Thus, we have that

$$
\mathrm{Var}(\|s\|^2) = 2\frac{1}{cM}
$$

$\square$

**Lemma 15.** $\|t\|^2$ *term.*

*Proof.* First, we note that $t$ only appears when $M < N_{trn}$. Thus, we only need to deal with that case. For this term, we note that $(I - A_{trn}^\dagger A_{trn})$ is a projection matrix onto a uniformly random $N_{trn} - M$ dimensional subspace. Then similar to $\|s\|^2$, we have that

$$
\mathbb{E}[\|t\|^2] = \mathbb{E}[v_{trn}^T v_{trn} - v_{trn}^T A_{trn}^\dagger A_{trn} v_{trn}] = \mathbb{E}\left[1 - a^T \begin{bmatrix} I_M & 0 \\ 0 & 0 \end{bmatrix} a\right] = 1 - \sum_{i=1}^{M} \frac{1}{N_{trn}} = 1 - c
$$

Similarly, we have that

$$
\begin{aligned}
\|t\|^4 &= \left(1 - \sum_{i=1}^{M} a_i^2\right)^2 \\
&= 1 + \left(\sum_{i=1}^{M} a_i^2\right)^2 - 2 \sum_{i=1}^{M} a_i^2 \\
&= 1 + \sum_{i=1}^{M} a_i^4 + \sum_{i \neq j}^{M} a_i^2 a_j^2 - 2 \sum_{i=1}^{M} a_i^2
\end{aligned}
$$

Taking the expectation, we get that

$$
\begin{aligned}
\mathbb{E}[\|t\|^4] &= 1 + 3 \sum_{i=1}^{M} \frac{1}{N_{trn}^2} + \sum_{i \neq j}^{M} \frac{1}{N_{trn}^2} - 2 \sum_{i=1}^{M} \frac{1}{N_{trn}} \\
&= 1 + \frac{3c}{N_{trn}} + \frac{N_{trn}(N_{trn}-1)}{M^2} - 2c \\
&= 1 + \frac{3c}{N_{trn}} + c^2 - \frac{c}{N_{trn}} - 2c \\
&= (1-c)^2 + \frac{2}{cM}
\end{aligned}
$$

Thus, we have that

$$
\mathrm{Var}(\|t\|^2) = 2\frac{c}{N_{trn}}
$$

$\square$

Now we could just use the the fact that $|\mathbb{E}[XY] - \mathbb{E}[X]\mathbb{E}[Y]| < \sqrt{\mathrm{Var}(X)\mathrm{Var}(Y)}$. Another way to do this is via using big $O$ in probability. Which is defined as follows:

**Definition 1.** *We save that a sequence of random variables $X_n$ is $O_P(a_n)$, if there exists an $N$ such that for all $\epsilon > 0$, there exists a constant $L$ such that for all $n \geq N$, we have that $\Pr[|X_n| > La_n] < \epsilon$.*

Then the trace terms.

**Lemma 8.** *Under standard noise assumptions, we have that*

$$\mathbb{E}[Tr(h^T k^T A_{trn}^\dagger)] = 0$$

*and*

$$\mathrm{Var}(Tr(h^T k^T A_{trn}^\dagger)) = \chi_3(c)/N_{trn},$$

*where $\chi_3(c) = \mathbb{E}[1/\lambda^3]$, $\lambda$ is an eigenvalue for $AA^T$ and $A$ is as in Lemma 6.*

*Proof.* First we note that

$$\mathrm{Tr}(h^T k^T A_{trn}^\dagger) = \mathrm{Tr}((A_{trn}^\dagger)^T v_{trn} u^T (A_{trn}^\dagger)^T A_{trn}^\dagger) = u^T (A_{trn}^\dagger)^T (A_{trn}^\dagger A_{trn}^\dagger)^T v_{trn}).$$

Again let $A_{trn} = U\Sigma V^T$ be the SVD. Then, we have the middle terms depending on $A_{trn}$ simplifies to

$$(A_{trn}^\dagger)^T A_{trn}^\dagger (A_{trn}^\dagger)^T = U(\Sigma^\dagger)^T \Sigma^\dagger (\Sigma^\dagger)^T V^T.$$

Thus, again letting $b = u^T U$ and $a = V^T v_{trn}$. We see that

$$\mathrm{Tr}(h^T k^T A_{trn}^\dagger) = \sum_{i=1}^{M} a_i b_i \frac{1}{\sigma_i^3}.$$

Now if take the expectation, since $a, b$ are independent and mean 0, we see that

$$\mathbb{E}_{A_{trn}}[\mathrm{Tr}(h^T k^T A_{trn}^\dagger)] = 0.$$

Let us also compute the variance. Here we have that

$$\mathbb{E}[\mathrm{Tr}(h^T k^T A_{trn}^\dagger)^2] = \sum_{i=1}^{M} \mathbb{E}\left[\frac{1}{\sigma_i^6}\right] \mathbb{E}[a_i^2]\mathbb{E}[b_i^2] + 0.$$

Now for the Marchenko Pastur distribution we have that the expectation of $1/\lambda^3 = \chi_3(c)$. where $\chi_3$ is some function. Thus, we have that

$$\mathbb{E}[\mathrm{Tr}(h^T k^T A_{trn}^\dagger)^2] = \frac{1}{N_{trn}}\chi_3(c) + o(1).$$

$\square$

**Lemma 9.** *Under standard noise assumptions, we have that*

$$Tr((A_{trn}^\dagger)^T kk^T A_{trn}^\dagger) = \frac{c^2}{(1-c)^3} + o(1)$$

*and*

$$\mathrm{Var}(Tr((A_{trn}^\dagger)^T kk^T A_{trn}^\dagger)) = \frac{3}{M}\chi_4(c) - \frac{1}{M}\frac{c^4}{(1-c)^6}$$

*where $\chi_4(c) = \mathbb{E}[1/\lambda^4]$, $\lambda$ is an eigenvalue for $AA^T$ and $A$ is as in Lemma 6.*

*Proof.* Now using Lemma 6, we see that

$$\mathbb{E}_{A_{trn}}[\mathrm{Tr}((A_{trn}^\dagger)^T kk^T A_{trn}^\dagger)] = \frac{c^2}{(1-c)^3}.$$

Similar to proofs before, we have that

$$\mathbb{E}_{A_{trn}}[\mathrm{Tr}((A_{trn}^\dagger)^T kk^T A_{trn}^\dagger)^2] = \sum_{i=1}^{M} \frac{3}{M^2}\chi_4(c) + \sum_{i\neq j} \frac{1}{M^2}\frac{c^4}{(1-c)^6} + o(1).$$

Where $\chi_4(c) = \mathbb{E}[1/\lambda^4]$ for the Marchenko Pastur distribution. Thus, we have that

$$\mathrm{Var}(\mathrm{Tr}((A_{trn}^\dagger)^T k k^T A_{trn}^\dagger)) = \frac{3}{M}\chi_4(c) + \frac{1}{M}\frac{c^4}{(1-c)^6} + o(1).$$

$\square$

**Lemma 10.** *Under the same assumptions as Proposition 2, we have that* $Tr(h^T s^T) = 0$.

*Proof.* Here we note that $h^T = (A_{trn}^\dagger)^T v_{trn}$ and $s^T = u^T(I - A_{trn}A_{trn}^\dagger)^T$. Thus, we have that

$$\begin{aligned}
\mathrm{Tr}(h^T s^T) &= \mathrm{Tr}((A_{trn}^\dagger)^T v_{trn} u^T - (A_{trn}^\dagger)^T v_{trn} u^T (A_{trn} A_{trn}^\dagger)^T) \\
&= \mathrm{Tr}(v_{trn}^T A_{trn}^\dagger u) - \mathrm{Tr}(u^T (A_{trn} A_{trn}^\dagger)^T (A_{trn}^\dagger)^T v_{trn}) \\
&= \mathrm{Tr}(v_{trn}^T A_{trn}^\dagger u) - \mathrm{Tr}(v_{trn}^T A_{trn}^\dagger A_{trn} A_{trn}^\dagger u) \\
&= \mathrm{Tr}(v_{trn}^T A_{trn}^\dagger u) - \mathrm{Tr}(v_{trn}^T A_{trn}^\dagger u) \\
&= 0
\end{aligned}$$

$\square$

As we can see that if we take the expectation of $\|W\|$ over $A_{trn}$, since the variance of each of the terms is small, we can approximate $\mathbb{E}[XY]$ with $\mathbb{E}[X]\mathbb{E}[Y]$. Then we get the following.

If $M < N_{trn}$, we have that

$$\begin{aligned}
\mathbb{E}_{A_{trn}}[\|W\|^2] &= \frac{\theta_{trn}^2}{(1+\theta_{trn}^2 c)^2}\frac{c^2}{(1-c)} + 0 + \frac{\theta_{trn}^4(1-c)^2}{(1+\theta_{trn}^2 c)^2}\frac{c^2}{(1-c)^3} \\
&= c^2 \frac{\theta_{trn}^2 + \theta_{trn}^4}{(1+\theta_{trn}^2 c)^2(1-c)}.
\end{aligned}$$

On the other hand, $M > N_{trn}$, we have that

$$\begin{aligned}
\mathbb{E}_{A_{trn}}[\|W\|^2] &= \frac{\theta_{trn}^2}{(1+\theta_{trn}^2)^2}\frac{c}{c-1} + \frac{\theta_{trn}^4}{(1+\theta_{trn}^2)^2}\frac{c^2}{(c-1)^2}\frac{c-1}{c} \\
&= \frac{c}{c-1}\frac{\theta_{trn}^2(1+\theta_{trn}^2)}{(1+\theta_{trn}^2)^2} \\
&= \frac{\theta_{trn}^2}{1+\theta_{trn}^2}\frac{c}{c-1}.
\end{aligned}$$

Now combining everything together, we get that

$$\mathbb{E}_{A_{trn}, A_{tst}}\left[\frac{\|X_{tst} - W(X_t st + A_{tst})\|}{N_{tst}}\right] = \begin{cases} \frac{\theta_{tst}^2}{N_{tst}(1+\theta_{trn}^2 c)^2} + \frac{1}{M}c^2\frac{\theta_{trn}^2+\theta_{trn}^4}{(1+\theta_{trn}^2 c)^2(1-c)} & c < 1 \\ \frac{\theta_{tst}^2}{N_{tst}(1+\theta_{trn}^2 c)^2} + \frac{1}{M}\frac{\theta_{trn}^2}{1+\theta_{trn}^2}\frac{c}{c-1} & c > 1 \end{cases}.$$

### B.5 PROOF OF THEOREM

We can see that the main text has how to put all of the pieces together to prove the main Theorem. We don't replicate that here.

### B.6 FORMULA FOR $\hat{\theta}_{opt-trn}$

As stated in the main text, we only need to take the derivative. So, we don't present that calculation here as it is fairly straightforward.

## C GENERALIZATIONS

In this section we discuss some possible generalizations of the method.

## C.1 HIGHER RANK

Let us present some heuristics for the higher rank formula. To do so we shall need some notation. Let $X_{trn} = \sum_{i=1}^{r} \sigma_i^{trn} u_i (v_i^{trn})^T$. Let $A$ be the noise matrix. Then for $1 \leq j \leq r$, define

$$A_j = \left( A + \sum_{i=1}^{j-1} \sigma_i^{trn} u_i (v_i^{trn})^T \right)$$

We shall now make some assumptions. Specifically, we assume that $u_j, v_j^{trn}$, and $A_j$ are all such that for $i_1 \neq i_2$, and for all $j$ we have that

$$\mathbb{E}[u_{i_1}^T A_j A_j^\dagger u_{i_2}] = \mathbb{E}[(v_{i_1}^{trn})^T A_j^\dagger A_j v_{i_2}^{trn}] = 0.$$

Additionally, we assume that for all $i_1, i_2, j$ we have that $\mathbb{E}[(v_{i_1}^{trn})^T A_j^\dagger u_{i_2}] = 0$. We also assume that the variance of these terms goes to 0 as $N_{trn}, M$ go to infinity.

**Lemma 16.** *With the given assumptions, we have that for all $i < j$,*

$$\sigma_i^{trn} u_i (v_i^{trn})^T A_j^\dagger \approx \sigma_i^{trn} u_i (v_i^{trn})^T A_{j-1}^\dagger \approx \sigma_i^{trn} u_i (v_i^{trn})^T A_{j-2}^\dagger \approx \ldots \approx \sigma_i^{trn} u_i (v_i^{trn})^T A_{i+1}^\dagger$$

*Proof.* Write $A_j = A_{j-1} + \sigma_j^{trn} u_j (v_k^{trn})^T$ and the use Meyer (1973) to expand the pseudoinverse of $A_j$. When we do this, we see that due to the assumption all terms expect $\sigma_i^{trn} u_i (v_i^{trn})^T A_{j-1}^\dagger$ are small. $\qquad \square$

Define $h_j = (v_j^{trn})^T A_j^\dagger$, $k_j = \sigma_j^{trn} A_j^\dagger u_j$, $t_j = (v_j^{trn})^T (I - A_j^\dagger A_j)$, $s_j = \sigma_j^{trn}(I - A_j A_j^\dagger) u_j$, $\beta_j = 1 + \sigma_j^{trn}(v_j^{trn})^T A_j^\dagger u_j$, $\tau_1^{(j)} = \|t_j\|^2 \|k_j\|^2 + \beta_j^2$, $\tau_2^{(j)} = \|s_j\|^2 \|h_j\|^2 + \beta_j^2$, and similarly $p_1^{(j)}, p_2^{(j)}, q_1^{(j)}$, and $q_2^{(j)}$. Now, we can write

$$X_{trn} + A = \sigma_r^{trn} u_r (v_r^{trn})^T + A_{r-1}$$

Then we have that

$$W = X(\sigma_r^{trn} u_r (v_r^{trn})^T + A_r)^\dagger = \sum_{i=1}^{r} \sigma_i^{trn} u_i (v_i^{trn})^T (\sigma_r^{trn} u_r (v_r^{trn})^T + A_r)^\dagger$$

Expanding and using the lemma, we get that

$$W \approx \sum_{i=1}^{r} \sigma_i^{trn} u_i (v_i^{trn})^T A_{i+1}^\dagger = \begin{cases} \sum_{i=1}^{r} \frac{\sigma_i^{trn} \beta_i}{\tau_1^{(i)}} u_i h_i + \frac{(\sigma_i^{trn})^2 \|t_i\|^2}{\tau_1^{(i)}} u_i k_i^T A_i^\dagger & c < 1 \\ \sum_{i=1}^{r} \frac{\sigma_i^{trn} \beta_i}{\tau_2^{(i)}} u_i h_i + \frac{(\sigma_i^{trn})^2 \|h_i\|^2}{\tau_2^{(i)}} u_i s_i^T & c > 1 \end{cases}$$

Where the second equality comes from the rank 1 results.

Now that we have an approximation for $W$ (given our assumptions), we can now approximate the variance and bias terms again. Let $W_i$ denote the $i$th factor (corresponding to $u_i$) of $W$. First, for the bias, due to the orthogonality of the $u$'s we get that

$$\|X_{tst} - W X_{tst}\|_F^2 = \sum_{i=1}^{r} \left\| \sigma_i^{tst} u_i (v_i^{tst})^T - W_i \sum_{j=1}^{r} \sigma_i^{tst} u_i (v_i^{tst})^T \right\|_F^2$$

Again, using our assumptions, we see that the terms in the $j$ summation dropout besides when $j = i$. Then again using our rank 1 result, we get that

$$\|X_{tst} - W X_{tst}\|_F^2 = \sum_{i=1}^{r} \left( \frac{\beta_i}{\tau_{idx}^{(i)}} \sigma_i^{tst} \right)^2$$

For the variance, we again estimate the norm of $W$ by expanding the trace. Here we see that the cross terms are 0 due to factors of $u_{i_1}^T u_{i_2}$. For the diagonal terms, we again use the rank 1 results and get that

$$\|W\|_F^2 = \sum_{i=1}^{r} \frac{(\sigma_i^{trn})^2 \beta_i^2}{(\tau_1^{(i)})^2} \text{Tr}(h_i^T h_i) + 2 \frac{(\sigma_i^{trn})^3 \|t_i\|^2 \beta_i}{(\tau_1^{(i)})^2} \text{Tr}(h_i^T k_i^T A_i^\dagger) + \frac{(\sigma_i^{trn})^4 \|t_i\|^4}{(\tau_1^{(i)})^2} \text{Tr}((A_i^\dagger)^T k_i k_i^T A_i^\dagger)$$

and if $c > 1$, then we have that

$$\|W\|_F^2 = \sum_{i=1}^{r} \frac{(\sigma_i^{trn})^2 \beta_i^2}{(\tau_2^{(i)})^2} \mathrm{Tr}(h_i^T h_i) + 2\frac{(\sigma_i^{trn})^3 \|h_i\|^2 \beta_i}{(\tau_2^{(i)})^2} \mathrm{Tr}(h_i^T s_i^T) + \frac{(\sigma_i^{trn})^4 \|h_i\|^4}{(\tau_2^{(i)})^2} \mathrm{Tr}(s_i s_i^T).$$

The final step would be to estimate each of these terms using random matrix theory. However, unfortunately the $A_j$ may not satisfy all of the needed conditions. However, we know that $A_j$ is a perturbation of $A$ and $A$ satisfies all of the needed conditions. Hence, if the perturbation is small, we can replace $A_j$ with $A$ and hopefully not incur too much cost. Note this is also the reason why the previous assumptions might be reasonable. If we replace $A_j$'s with $A$ use our estimates from the rank 1 result. We then get our estimate for the generalization error for general rank $r$ data.

$$R(\theta_{trn}, \theta_{tst}, c, \Sigma_{trn}, \Sigma_{tst}) = \sum_{i=1}^{r} \frac{(\theta_{tst}\sigma_i^{tst})^2}{N_{tst}(1 + (\theta_{trn}\sigma_i^{trn})^2 c)^2} + \frac{c^2((\theta_{trn}\sigma_i^{trn})^2 + (\theta_{trn}\sigma_i^{trn})^4)}{M(1 + (\theta_{trn}\sigma_i^{trn})^2 c)^2(1 - c)} + o(1) \tag{12}$$

and if $c > 1$, we have that

$$R(\theta_{trn}, \theta_{tst}, c, \Sigma_{trn}, \Sigma_{tst}) = \sum_{i=1}^{r} \frac{(\theta_{tst}\sigma_i^{tst})^2}{N_{tst}(1 + (\theta_{trn}\sigma_i^{trn})^2)^2} + \frac{c(\theta_{trn}\sigma_i^{trn})^2}{M(1 + (\theta_{trn}\sigma_i^{trn})^2)(c - 1)} + o(1). \tag{13}$$

In the experimental section, we see that for small values of $r$ for $c$ bounded away from 1. This seems to be good estimate for the generalization error.

### C.2 RANDOM FEATURES MODEL

Here we assumed that our data is given $X = U\Sigma V^T$. One generalization of this that we have a $G$ who entries i.i.d Gaussian, or whose columns are uniformly distributed on the unit sphere. Then for non linear function $\sigma$, we assume that $X = \sigma(GU\Sigma V^T)$. If we assume that $\sigma$ is linear, the we have that as $L \to \infty$, $G^T G \to I_M$. Thus, we have that $GU$ approximately satisfies our assumptions of orthogonal columns. Hence we expect our formula to still be a reasonable approximation.

## D EXPERIMENTS

Please see accompanying notebook for code to produce the data for all of the figures.

### D.1 LOW SNR AND HIGH SNR DATA

For low SNR data, we sample the $\theta$ times singular values from a squared standard Gaussian. We do this independently for all $2r$ singular values. We call this the low SNR region because $\theta$ is not being scaled with the number of data points. Hence as $N_{trn}, N_{tst} \to \infty$, the SNR goes to 0.

For the high rank data, we sample $\theta$ times singular values from a squared Gaussian and then multiply by $\sqrt{N_{trn}}, \sqrt{N_{tst}}$. Hence here the SNR does not go to 0 as $N_{trn}, N_{tst} \to \infty$.

# E    GENERALIZATION ERROR VERSUS TRAINING NOISE LEVEL PLOTS

## E.1    MORE TESTS FOR RANK 1

Here we provide more examples of $c$ and how our theoretical formula matches the experimental performance exactly.

Each empirical point is the average over 50 trials. These were run on a laptop with 8gb of RAM and an i3 processors. The average time to produce any of these plots is about 10 to 30 minutes.

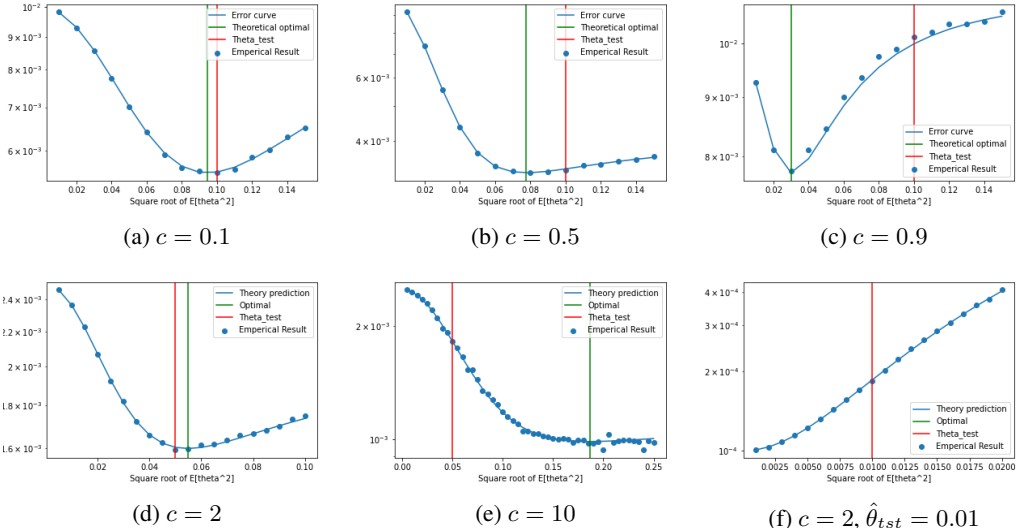

(a) $c = 0.1$        (b) $c = 0.5$        (c) $c = 0.9$

(d) $c = 2$        (e) $c = 10$        (f) $c = 2, \hat{\theta}_{tst} = 0.01$

Figure 8: Figures (a) - (e) showing the accuracy of the formula for the expected mean squared error for $c = 0.1, 0.5, 0.9, 2, 10$ for fixed value of $\hat{\theta}_{tst}$. Figure (f) empirically verifies the existence of a regime where training on pure noise is optimal. Here the red and green lines represent $\mathbb{E}[\hat{\theta}_{tst}^2]$ and $\mathbb{E}[\hat{\theta}_{trn}^2]$ respectively. Each empirical data point is averaged over at least 50 trials.

## E.2    RANK 2 DATA

Let us now demonstrate that the double descent shaped curve exists beyond rank 1 data and linear autoencoders. We will do this by gradually making the set up more complicated until we can no longer recreate this phenomena. First, we consider rank 2 data is of the following form. Let $W_{data}$ be some fixed matrix, then our data is generated by

$$X = \texttt{relu}(W_{data}\texttt{relu}(uv^T).$$

Where a different $v$ is sampled for the training and test data. the results for this can be seen in Figure 9. As we can from the figure, we have the exact same qualitative trend for $c$ that we saw before. That is, as $c$ goes from 0 to 1, we have that $\hat{\theta}_{trn}$ goes from $\hat{\theta}_{tst}$ to 0, and then as $c \to \infty$, we have that $\hat{\theta}_{trn}$ goes to infinity as well.

## E.3    MNIST DATA

We now look at the linear network with MNIST data.

### E.3.1    NON-LINEAR NETWORK

Here, we trained each network for 1500 epochs. During each epoch we computed a gradient using the whole data set. We used Adam as the optimizer with the code written in Pytorch. Each data point was generated over 20 trials.

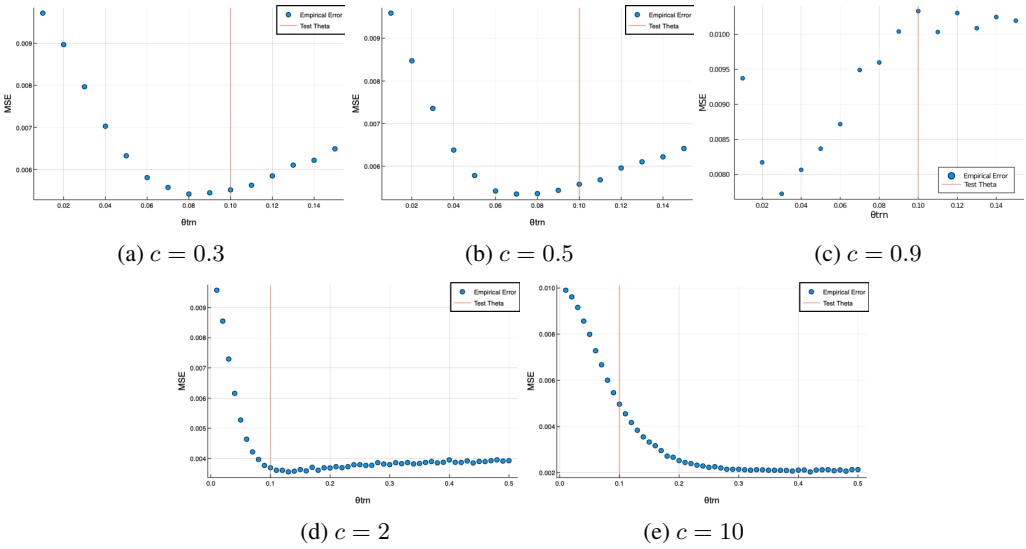

Figure 9: Rank 2

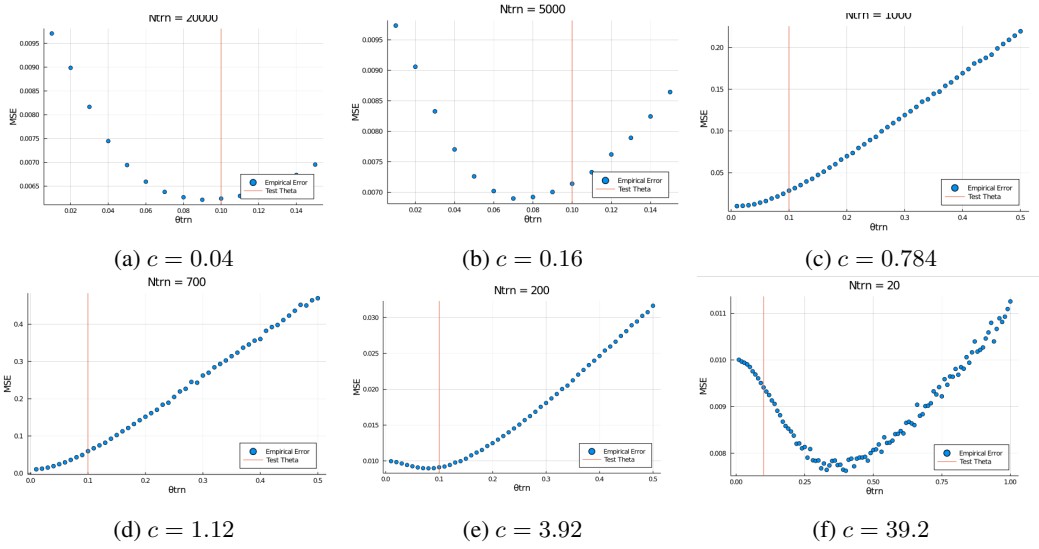

Figure 10: MNIST

These experiments take a little bit more time to run and the one with bigger amounts of data can take upto 5 hours on a google cloud instance with 16gb RAM. Here we used a Telse P4 gpu.

LRL - is a model with a reLU at the end of the first layer only.

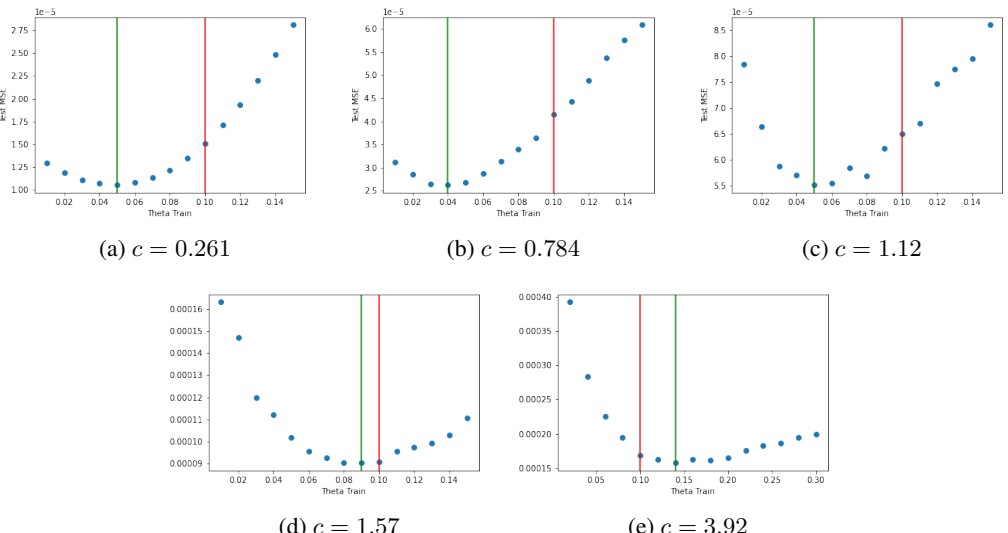

(a) $c = 0.261$

(b) $c = 0.784$

(c) $c = 1.12$

(d) $c = 1.57$

(e) $c = 3.92$

Figure 11: MNIST - LRL model

