# OpenReview forum: "Training Data Size Induced Double Descent For Denoising Neural Networks and the Role of Training Noise Level"
_ICLR.cc/2022/Conference — ICLR 2022 Submitted_

### Official Review · Reviewer_dPcd · 2021-11-02

**Correctness:** 3
**Technical Novelty And Significance:** 2
**Empirical Novelty And Significance:** 4
**Recommendation:** 6
**Confidence:** 3

**Main Review:**

I think the topic of this paper is interesting. I like the result that the optimal amount of noise in the training data is not equal to the noise level in the test data. The following are some detailed comments.

1. Authors claim that the noise in the training samples can be controlled. That is misleading. Actually, we can enlarge the noise by manually adding more noise into the training data, but we cannot reduce the noise. A related question is whether Eq.(7) suggests that we can actually get a better result by adding more noise to the training data.

2. The explanation of the double descent with respect to the number of training samples is not satisfactory. Eq.(3) and (4) only suggest that the error has a peak when $c=1$, i.e., when $N_{trn}=M$. It only implies that when N_trn increases, the error will increase to the peak in the region $N_{trn}\in (M-{small\ value},M)$ and then decrease when $N_{trn}\in (M, M+{small\ value})$. Maybe it will be clear to see the monotonicity by doing the derivative of Eq.(3) and (4) with respect to $N_{trn}$.

3. How does Theorem 1 compare with results about double descent in other literature? For example, is the descending speed with respect to the number of training samples characterized by Theorem 1 similar to that in the existing literature?

4. In Eqs. (3)~(7), N_tst, M, and c appear at the same time. Since c=M/N_trn, maybe it is a good idea to replace M with c*N_trn.

**Summary Of The Paper:**

This paper considers a denoising setup for a linear model. Compared with the traditional supervised learning setup where the noise is added to the output, the denoising setup has noise in the input. The authors provide an asymptotically exact formula for the generalization error for rank 1 data and an approximation for higher rank data. Both the theoretical and numerical results validate that the generalization error has a peak when the number of samples is equal to the number of features, which is consistent with existing literature about the double descent phenomenon. Authors also derive a formula for the amount of noise in the training data for the best generalization performance, which leads to a novel conclusion that the optimal noise amount in the training data is not always equal to the noise level in the test data.

**Summary Of The Review:**

This paper is a good paper with novel results in the denoising setup, yet some places can be made clearer.

---

> ### Author Response · Authors · 2021-11-10
> **Answering Questions**
>
> We thank the reviewer for the helpful comments. We hope that the following clarifications increases the reviewers enthusiasm for our work.
>
> 1) Yes for real world examples we can only add noise. Eq (7) actually does suggest that we want to remove noise. However, this is only the case in when we extremely few data samples. This can be seen in Figure 5b of the revision. Where for large $c$, (i.e. small $N_{trn}$) we have that the ratio dips below 1. On the other hand, in the infinite data sample case, it suggest that train SNR = test SNR.
>
> 2) This true, we do not show monotonicity by looking at the derivative. But we do need to look at the derivative. Let us assume that $\theta\_{trn} = \hat{\theta}\_{trn}\sqrt{N\_{trn}}$, that $M$ is fixed and $c$ changes by changing $N_{trn}$.
>
> Then, let us look at equation (3). Here we see that as $c$ increases to 1, the bias stays constant as $\theta_{trn}^2c = \hat{\theta}\_{trn}^2M$.
>
> Similarly, we see that the variance terms looks likes
>
> $$\frac{1}{(1+\hat{\theta}\_{trn}^2M)^2)}\left(\frac{\hat{\theta}\_{trn}^2c}{(1-c)} + \frac{M\hat{\theta}\_{trn}^4}{1-c}\right)$$.
>
> Thus, the variance monotonically increases as $c \to 1$.
>
> If we look at equation (4), we see that the bias is monotonically decreasing and the variance now looks like $\frac{1}{c-1}$ and hence changes monotonically for $c > 1$ (which is when the formula applies).
>
> 3) This descent speed is similar to [1]. Look at Theorem 1. Their $\gamma$ is the same as our $c$.
>
> 4) In our paper, we want to think of $M$ as a constant. Hence I think leaving it as $M$ might be easier to parse the asymptotics.
>
>
> [1] T. Hastie, A. Montanari, S. Rosset, and R. Tibshirani. Surprises in high-dimensional ridgeless least squares interpolation. ArXiv, abs/1903.08560, 2019.

---

> > ### Comment · Reviewer_dPcd · 2021-11-29
> > **Re: Answering Questions**
> >
> > Thanks for your response. I will keep my original score.

---

### Official Review · Reviewer_5V96 · 2021-11-02

**Correctness:** 3
**Technical Novelty And Significance:** 4
**Empirical Novelty And Significance:** 2
**Recommendation:** 6
**Confidence:** 4

**Main Review:**

**Writing and flow:**

Overall the writing and flow of the paper are good. What is being done is clear and makes sense. This is great and a strength.

**Assumptions and model:**

The assumptions (e.g., noise statistics) and model used are all reasonable (strength).

**Introduction:**

The introduction has good logic flow and is easy to understand the intuition and motivation behind the paper, which is a strength. The authors do a good job explaining their points. I especially found it interesting the idea that, if double descent depends on the number of samples, that datasets may be unfortunately placed at the peak without our knowing.

Now for a few weaknesses. The intuition that noise regularizes is useful but not absolutely accurate. If I am not mistaken, please correct me if I am, additive noise only regularizes in a supervised setting with small noise amplitude (Bishop, 1995). Thus, the claim, which is made in the first paragraph of page 3, that more regularization via adding noise doesn’t remove the double descent shape is a weak claim because adding lots of noise may not actually be equivalent to regularization.

Also, the figure labels on Figure 3 are extremely confusing and the axis labels are so small that I can’t really make out what Figure 3 is actually saying. Figure 4a is good but Figure 4b is confusing. The caption says that the plot is of training SNR and the axis label says it is a ratio instead.

**Related work:**

The related work section is very small and needs to be expanded in order put the author’s contribution into context. This is a weakness.

**Theorem 1:**

Theorem 1 is great and the section is well written. The only confusing part for me is how “The $o(1)$ error term goes to $0$ as $N_{trn}, M \rightarrow \infty.$”? This mean that the error term depends on $N_{trn}, M$ and hence is not constant, i.e., $o(1)$.

**Section 3.1:**

The implications of Equation 7 are very interesting to me and I think this is a great contribution because it is non-intuitive.

**Section 3.2:**

As mentioned above, saying that you have “optimal regularization” because you pick the optimal amount of training noise does not make sense because they are not equivalent if the noise variance is too large. In fact, you can actually see no double descent if you just set the noise variance to 0. This is a weakness of the paper.

**Experiment and conclusion:**

The experiment section is very short and there is no conclusion, which makes the paper incomplete. The conclusion section is important but most important is the experiment section. The results for MNIST are quite interesting actually but there isn’t much detail in the paper describing them, which is unfortunate. There is some information in the appendix, but it should be described in the original paper. Some of the figures can be removed to make room for it, in my opinion, such as Figure 1 and 6.

**Summary Of The Paper:**

In this paper, the authors study overparameterization of autoencoders in terms of varying sample size. They derive an analytic expression for the bias-variance decomposition of the loss for a bottleneck of size 1. Furthermore, they approximate the bias-variance decomposition for larger bottleneck sizes. This is done with all linear networks. They experimentally show double descent in nonlinear networks as well.

**Summary Of The Review:**

**Strengths:**

The paper shows a novel result: the bias-variance decomposition of the rank-1 linear denoising autoencoder. This introduces some interesting new concepts, such as optimal training SNR as compared to test SNR. The paper is clearly written and is easy to understand. In addition, the authors find an approximation to the bias-variance decomposition for bottlenecks of larger size than 1.

**Weaknesses:**

The others equate adding noise and regularizing the model, which is not equivalent if the noise variance is too large. If they want to make statements about regularization, then they should explicitly regularize and see what happens. If the results are similar then even that is convincing, even if it is empirical. Moreover, some of their figures are hard to read and interpret. Finally, the authors leave out most of the details regarding the experiments and completely omit the conclusion.

Overall, I based my score on the fact that the paper is incomplete and rough, not because the work has major problems. This is very interesting work and should be polished before next submission.

---

> ### Author Response · Authors · 2021-11-10
> **Regularization and rewriting**
>
> We thank the reviewer for their detailed comments. We hope that our revision and the following discussion removes any lingering doubts.
>
> Please see the comment on rewriting for the various edits to the paper.
>
> To address the point on regularization. We have removed the claims on noise as a regularizer and now only refer to it as the variance of the noise and its effects on double descent.
>
> However, I do think in our case increasing noise variance is equivalent to increasing the regularization.
>
> The problem that we solve is $\|\|\theta X - W(\theta X + A)\|\|$ where $A$ is the noise matrix. To highlight, the noise variance, we can rewrite the problem as minimizing $\|\|X - W(X - \frac{1}{\theta}A)\|\|$. Note these are equivalent due to the following.
>
> $$W_{opt} = \theta X (\theta X + A)^\dag = \theta X (\theta (X + \frac{1}{\theta}A))^\dag = \theta \frac{1}{\theta} X (X+\frac{1}{\theta}A)^\dag = X (X+\frac{1}{\theta}A)^\dag.$$
>
> Where $W_{opt}$ is the optimal $W$. Now for rank 1 data, we can use our formulas from the paper to see how the norm of $W_{opt}$ changes as a function of $\theta$. If $\sigma = 1$ is the one singular value then we see that (the value of sigma will not effect the result)
>
> $$\mathbb{E}[\|\|W_{opt}\|\|_F^2] = \begin{cases} \frac{c^2(\theta^2+\theta^4)}{M(1+\theta^2c)^2(1-c)} & c < 1 \\\\ \frac{c\theta^2}{(M(1+\theta^2)(c-1)} & c > 1 \end{cases}. $$
>
> In both cases, we see that as $\theta \to 0$ (i.e. the variance goes to infinity), we have that $\mathbb{E}[\|\|W_{opt}\|\|_F^2] \to 0$. Thus, even for large variance, I think it is okay to think of it as a regularizer.
>
> ----------
>
> Suppose we change the problem slightly. In our problem, we first compute $W_{opt}$ and then compute the expectation. We can flip the order, that compute $W^*$ that minimize
>
> $$\mathbb{E}[\|\|X - W(X+\frac{1}{\theta}A)\|\|_F^2].$$
>
> In this case (if the entries of $A$ have variance 1), using a similar proof to Lemma 1 and Lemma 3, we see that,
>
> $$\mathbb{E}[\|\|X - W(X+\frac{1}{\theta}A)\|\|_F^2] = \|\|X-WX\|\|_F^2 + \frac{1}{\theta^2}\|\|W\|\|_F^2.$$
>
> In this case, the noise variance and the regularization parameter are directly related.

---

> > ### Comment · Reviewer_5V96 · 2021-11-24
> > **Thank you for the edits**
> >
> > Thank you for adding a conclusion and adding details about the experiments. Your explanation about noise as a regularizer also makes sense and I appreciate the effort. Overall the paper is now much more polished and I will raise my score from a 3 to a 6. Thanks for all the good work!

---

### Official Review · Reviewer_yBGu · 2021-11-02

**Correctness:** 3
**Technical Novelty And Significance:** 3
**Empirical Novelty And Significance:** 3
**Recommendation:** 3
**Confidence:** 3

**Main Review:**

Strengths:
1. Double descent has been observed in various contexts and it's an interesting and important question to understand it thoroughly. While previous work only focused on supervised learning settings, this paper considers a denoising setting instead, which is a reasonable first step to consider in the unsupervised space.
2. The paper gives an answer in the rank-1 case in terms of the double descent phenomenon wrt the training data size. There is also some interesting characterization of the optimal SNR.

Weaknesses: Overall, this paper is not well written and needs significant improvement. There are a lot of things not explained clearly, and it's very difficult to follow the paper. Below are some major questions/issues.
1. What are the scales of all the relevant quantities in the main result (Thm. 1)? Which are considered constants and which grow with $N, M$? This needs to be explained clearly.
2. The data model setup also needs some more explanation. It's assumed that $V$ has orthonormal columns. Why is this a reasonable assumption? This assumption means that the datapoints are not independent, which makes it unrealistic for statistical learning.
3. The rank-r result is not explained clearly. It is claimed that formulas (5) and (6) are "approximations." In what sense are they approximating the true risk? What does it mean to "assume that $A_j$ has similar spectral properties to $A_{trn}$"?
4. Even the introduction section is quite chaotic. There's first some motivation and an overview of the type of results in the paper (which is itself somewhat confusingly written), and then suddenly at "contributions" it goes back to the motivation again...
5. At the bottom of page 6, it is claimed that picking the optimal training noise would not prevent double descent, in contrast to other results showing that optimal regularization prevents double descent. What about other regularization? This doesn't seem to rule out the possibility that other regularization can eliminate double descent.

Minor points:
1. It is claimed (page 3) that a 2-layer linear network model is considered. However, the model is just a usual (1-layer) linear model, not a linear network.
2. In the setup (Section 2 beginning), $V\in\mathbb{R}^{r\times N}$ should be $V\in\mathbb{R}^{N\times r}$?
3. In "contributions" (page 3): ".... focus on representation learning..." I don't think there's any representation learning aspect here.

**Summary Of The Paper:**

This paper identifies and studies a double descent phenomenon (wrt the training data size) for a denoising problem, where the goal is to recover a low-rank ground truth given a corrupted version of it (with added full-rank noise). The paper considers the linear regression estimator, and gives the exact formula for the test risk in the asymptotic regime for the rank-1 case; for a general rank r, the paper also gives a formula, but it's not entirely rigorous. Furthermore, the paper also studies the role of the training and testing SNRs.

**Summary Of The Review:**

This paper extends double descent (wrt data size) to a low-rank denoising setting with linear regression. While the result is novel and has some interesting aspects, many aspects of the problem and the results are not clearly explained, and the writing overall needs significant improvement.

---

> ### Author Response · Authors · 2021-11-10
> **Revision based on comments**
>
> We thank the reviewer for the insightful comments. We hope that the new revision and the following discussion, persuades the reviewer that we have addressed all of their concerns.
>
> 1) $\theta_{trn}$ is supposed to be $O(\sqrt{N_{trn}})$. The $\sigma_i$ are fixed. We added this to the paper.
>
> 2) We are not assuming that there is any distribution on our data points. Hence there isn't a notion of independence for the data samples. However, what we are assuming is that all of data lies on some low dimensional linear subspace $L$. That is, the space spanned by the columns of $U$. Now let $x_1, x_2, \ldots, x_n$ be **any** $n$ data points from $L$. Then let $X = [x_1 \ldots x_n]$. Then this matrix has rank at most $r$. Since the columns come from a rank $r$ subspace. Hence, if we look at the SVD of $X$ it has the form $U\Sigma V^T$ The left singular vectors are given by $U$ and the right singular vectors are orthogonal and the sigmas are the singular values for the matrix.
>
> 3) The gist is that we assume that the $A_j$s satisfy all of our noise assumptions. (The append has a more nuanced discussion, where if we assume less, we can get an a slightly weaker result)
>
> Hence the formula is only true in this very specific case. Thus, we claim that the formula is an approximation. In particular, experimentally, we see that formula is accurate for low rank data and low SNR data. This is because in those situations $A_j \approx A$ and hence the above assumption is closer to being true.
>
> 4) We have rewritten the introduction. Please see the revised version.
>
> 5) Yes, if we add $L_2$ regularization to our problem, it may be the case, that we can mitigate double descent. The remark on page 6 was in relation to how for various problems, like ridge regression, optimally setting the regularization constant mitigates double descent. To draw a parallel between that literature and our paper, we note that we do not have any explicit $L_2$ regularizer, but that setting noise variance is an analog. However, for us, setting the noise variance optimally, doesn't mitigate double descent.

---

> > ### Comment · Reviewer_yBGu · 2021-11-29
> > **Follow up**
> >
> > Thank you very much for the answers and revisions. These resolved some of the questions, but I am still not convinced that the result in this paper is significant enough due to some remaining concerns.
> >
> > First, regarding the data assumptions, I don't think it's true that there is no assumption except that they are in a low-dim subspace. In order for the comparison between different sample sizes (or different values of $c$) to be meaningful, you essentially need to fix the singular values $\sigma_i$ and then vary $c$. I'm not sure how this can be naturally satisfied or why this is a reasonable model. The reason I brought up iid data is that it's the most natural way to study the effect of varying sample size (and underlies all of ML).
> >
> > Second, the explanation for the higher rank case is still vague. There is no explanation when $A_j$ can satisfy the assumptions. The only example provided is when $A_j \approx A$, which is simply because the noise is of a much larger scale. The more general case is not addressed and it appears that the given formula isn't accurate so possibly some correction terms are needed.

---

> > > ### Author Response · Authors · 2021-11-29
> > > **Re: Follow up**
> > >
> > > Thank you for the follow up and for discussing our work with us.
> > >
> > > *"First, regarding the data assumption.. (and underlies all of ML)."*
> > >
> > > In relation to this point, our formula allows for the singular values of be changed as well. However, you are right, for most data, we would need some model on how the singular values change as we change for $c$. For the case of rank 1 data, we are assuming that the singular value scales as $\sqrt{N_{trn}}$. This is the reason we look at $\hat{\theta}\_{trn}$ and why formula (7) on page 5 gives a formula for $\theta_{trn}^2$ normalized by $N_{trn}$. In general, we picked this scaling so that the SNR stayed constant.
> > >
> > > **However, the point of the theoretical model was to explain empirical phenomena that we had noticed.** Specifically, the phenomena shown in Figures 3 and 4. Namely, that for a fixed test data and test SNR, the optimal training SNR is not equal to the test SNR and that this optimal SNR follows a double descent curve. **To the best of our knowledge, this phenomena had not been seen before and is a major contribution of our work.** To provide a theoretical model (even if it is not the most general model) to provide justification is the goal of our analysis.
> > >
> > > *"Second, the explanation for the higher rank case is still vague"*
> > >
> > > As the reviewer points out, this is the case, when we have a large amount of noise. The other case empirically where this seems to be true is when the rank is low compared to the number of data points, Figure 7 b (rank 2, 3, 5 cases the formula is fairly accurate). Hence if the number of data points go to infinity and the data all comes from the same low rank space, we hope the formula would hold.
> > >
> > > However, we do not believe the formula holds exactly as is for the general case. Hence we only think of it a heuristic approximation. We do not claim that this is exact formula and the reviewer points we are most likely missing some correction terms.

---

### Official Review · Reviewer_f7hC · 2021-11-03

**Correctness:** 3
**Technical Novelty And Significance:** 3
**Empirical Novelty And Significance:** 3
**Recommendation:** 5
**Confidence:** 2

**Main Review:**

Strengths: The paper investigates a novel double-descent phenomenon as a function of the number of samples for the denoising task. The authors make a large variety of experiments and investigate the phenomenon from different angles (i.e. neural networks, real-world datasets, etc).

Weaknesses: I found the paper structure confusing. The introduction takes 3.5 pages with many figures already presented before introducing the concepts. I would recommend moving Figure 1 or making it substantially smaller as the comparison between (a) and (b) can be made with such little effort (with some equations or a smaller figure). I was confused by what M is in Figures 3 and 4 which is introduced much later. The paper does not have a conclusion part.

Relevant work:
On page 2, I found that the citations are centered around Nakkiran et. al.'s work unfairly. Note that sample-wise double-descent is also studied in d'Ascoli et. al. 2019 (https://arxiv.org/pdf/2006.03509.pdf), Adlam et. al. 2020 (http://proceedings.mlr.press/v119/adlam20a). I see that these works are cited but I think these should be presented already on page 2 where sample-wise descent is motivated.

Moreover, for the double-descent and asymptotics of random features, the following works should be cited as well in the related works section on page 4: Jacot et. al. (http://proceedings.mlr.press/v119/jacot20a.html), Mel et. al. 2021 (https://proceedings.mlr.press/v139/mel21a.html).

Typos:
* page 4: U \Sigma V^T: T should be removed. also in many other places, there is some confusion on transposes.
* Figure 7 caption: for the our formula -> remove the

**Summary Of The Paper:**

The paper investigates a new double-descent in denoising task for linear predictors. As the number of samples increases the denoising loss exhibits a peak in the case of linear predictions, but the peaks are less pronounced in the case of neural networks, except for the case with rank 1 data. The authors calculate an asymptotical formula for their special rank-1 data model in theorem 1 which proves the explosion at (c=1). For the arbitrary rank data, the authors present an approximate formula too.

**Summary Of The Review:**

The paper proposes a low-rank data model and studies the double-descent phenomenon for denoising tasks. The main theorem presents an asymptotical formula for the risk for rank 1 data, and also approximate formulas for the general rank cases are presented. The theory is presented for the linear case, whereas the experiments cover also neural networks and MNIST.

---

> ### Author Response · Authors · 2021-11-10
> **Addressed Concerns**
>
> We thank the reviewer for the helpful comments. We hope that the following changes (as can be seen in the revision) removes any lingering doubts on the quality of the work.
>
> Please see the revised version of paper. Specifically, we
>
> 1) Removed the experiments from the intro and made it its own section.
> 2) Added a conclusion
> 3) No longer center on Nakkiran et al
> 4) Expanded the related works section
> 5) Added the relevant citations.
> 6) Fixed the typos

---

### Public Comment · ~Rishi_Sonthalia1 · 2023-04-17
**Accepted Version**

The accepted version at TMLR can be found at https://openreview.net/forum?id=FdMWtpVT1I

---

### Decision · Program_Chairs · 2022-01-20

**Decision:**

Reject

**Comment:**

This paper proposes a theory for double descent phenomena in denoting deep neural networks. There are two major concerns: (1) The assumption that the data lie in a low dimensional subspace is quite strong, and needs to be weaken or better justified. (2) The theory only works for r=1, where the rank is one. For general rank, how to apply the proposed analysis is hand wavy and not convincing. The paper can be significantly strengthen if these two issues could be addressed.